# Aurora-A mediated phosphorylation of LDHB promotes glycolysis and tumor progression by relieving the substrate-inhibition effect

Aoxing Cheng[1,7], Peng Zhang [1,7], Bo Wang[2,7], Dongdong Yang[1,7], Xiaotao Duan[2], Yongliang Jiang[1], Tian Xu[1], Ya Jiang[1], Jiahui Shi[1], Chengtao Ding[1], Gao Wu[1], Zhihong Sang[3], Qiang Wu[4], Hua Wang [5], Mian Wu [1], Zhiyong Zhang[1], Xin Pan [3], Yue-yin Pan[6], Ping Gao[1], Huafeng Zhang[1], Cong-zhao Zhou [1], Jing Guo [1]* & Zhenye Yang [1]*

Overexpressed Aurora-A kinase promotes tumor growth through various pathways, but whether Aurora-A is also involved in metabolic reprogramming-mediated cancer progression remains unknown. Here, we report that Aurora-A directly interacts with and phosphorylates lactate dehydrogenase B (LDHB), a subunit of the tetrameric enzyme LDH that catalyzes the interconversion between pyruvate and lactate. Aurora-A-mediated phosphorylation of LDHB serine 162 significantly increases its activity in reducing pyruvate to lactate, which efficiently promotes $NAD^+$ regeneration, glycolytic flux, lactate production and bio-synthesis with glycolytic intermediates. Mechanistically, LDHB serine 162 phosphorylation relieves its substrate inhibition effect by pyruvate, resulting in remarkable elevation in the conversions of pyruvate and NADH to lactate and $NAD^+$. Blocking S162 phosphorylation by expression of a LDHB-S162A mutant inhibited glycolysis and tumor growth in cancer cells and xenograft models. This study uncovers a function of Aurora-A in glycolytic modulation and a mechanism through which LDHB directly contributes to the Warburg effect.

---

[1] Hefei National Laboratory for Physical Sciences at Microscale, CAS key Laboratory of Innate Immunity and Chronic Disease, First Affiliated Hospital, Division of Life Sciences and Medicine, University of Science and Technology of China, Hefei, China. [2] State Key Laboratory of Toxicology and Medical Countermeasures, Beijing Institute of Pharmacology and Toxicology, Beijing, China. [3] Institute of Basic Medical Sciences, National Center of Biomedical Analysis, Beijing, China. [4] Department of Pathology, Anhui Medical University, Hefei, China. [5] Department of Oncology, the First Affiliated Hospital of Anhui Medical University, Hefei, China. [6] Department of Medical Oncology, the First Affiliated Hospital of USTC, Division of Life Sciences and Medicine, University of Science and Technology of China, Hefei, China. [7]These authors contributed equally: Aoxing Cheng, Peng Zhang, Bo Wang, Dongdong Yang. *email: jguo2013@ustc.edu.cn; zhenye@ustc.edu.cn

Extensive metabolic reprograming is a hallmark of malignant cancer cells[1]. Pyruvate, the glycolytic product, is transported into mitochondria to fuel the TCA cycle in normal epithelia. However, in tumor cells, most pyruvate is reduced to lactate in cytoplasm by lactate dehydrogenase (LDH), the enzyme catalyzes the interconversion between pyruvate and lactate[2–4]. This is known as the Warburg effect in cancers[4,5]. Tetrameric LDH comprises LDHA and LDHB, two subunits that are encoded by independent genes. In muscle or liver, most of the LDH is composed of four LDHA subunits, and preferably catalyzes the reduction of pyruvate to lactate. In heart and brain, LDH is mainly composed of four LDHB subunits, and predominantly catalyzes the oxidation of lactate to pyruvate[3,6]. In other tissues, LDH is composed of both LDHA and LDHB. Because the catalytic activity and direction of LDH control the rate of glycolysis[4], LDH is actively regulated, especially in cancer cells. Interestingly, most known regulations are conducted through the modulation of LDHA[7–9]. Although an essential role of LDHB in the progression of various cancers has been increasingly reported[10–13], how LDHB is precisely controlled in glycolytic regulation and tumor progression remains poorly understood.

Aurora-A, a conserved serine/threonine kinase, is responsible for centrosome maturation in G2 phase and bi-polar spindle formation in mitosis[14,15]. Aurora-A is also found to be over-expressed and/or amplified in various solid tumors[16,17]. High Aurora-A expression has been reported as an independent and significant prognostic biomarker in human cancers[16–18]. Aurora-A is activated through inter-molecular self-phosphorylation at threonine 288 in interphase[19,20], and it is also allosterically activated at spindle distal ends after binds with the partner TPX2 in mitosis[16,20]. Activated Aurora-A promotes tumor progression via multiple pathways including anti-apoptosis, metastasis, sustained proliferation and stemness[14,21,22]. However, whether Aurora-A also promotes tumor progression by directly modulating metabolic pathways remains unclear.

It has been reported that Aurora-A and p53 mutually inhibit each other. While p53 suppresses the transcription and promotes the degradation of Aurora-A, Aurora-A inhibits the transcriptional activity and reduces the stabilization of p53[16,23]. Given glycolytic level was proved to be negatively associated with the function of p53, it is possible that activated Aurora-A regulates glycolysis in p53-deficient cancer cells[2,24].

In this study, we find that Aurora-A directly interacts with and phosphorylates glycolytic enzyme LDHB. This modification increases its catalytic activity in the reduction of pyruvate to lactate, resulting in a boost in $NAD^+$ regeneration, glycolytic flux, and biosynthesis with glycolytic metabolites, which facilitate tumor progression. These results reveal a pathway by which Aurora-A promotes the Warburg effect upon post-translational modification of LDHB.

## Results

**Aurora-A promotes glycolysis in p53-deficient cancer cells.** To explore whether Aurora-A regulates metabolic reprograming, the glycolytic statuses were first analyzed in cancer cell lines with various activities of Aurora-A and p53. While the expression level and activity of Aurora-A are relatively low in A549, MCF-7, and RKO cells (p53 efficient), Aurora-A is considerably upregulated in DLD1, U251, and 293 T cells (p53-deficient) (Fig. 1a). The glycolytic status in these cells were then assessed. Both the levels and capacities of glycolysis were remarkably higher in DLD1, U251, and 293 T cells than in A549, MCF-7, and RKO cells (Fig. 1b), indicating that the expression levels of Aurora-A are proportional to the levels of glycolysis in these cells. Intriguingly, the glycolytic rates were markedly decreased when Aurora-A was

inhibited by the selective inhibitor MLN8237, and these decreases showed a time-dependent manner within four hours' treatment (Fig. 1c, d). Furthermore, when Aurora-A was inactivated by expressing kinase-dead (KD-) Aurora-A, a dominant-negative mutant[25] (Fig. 1e), the glycolytic rate was also markedly decreased (Fig. 1f). Consistently, the glycolysis was significantly decreased after Aurora-A inhibition in p53-deficient U251 and 293 T cells, but not in p53-efficient MCF-7 or RKO cells (Supplementary Fig. 1a–h). Together, these data demonstrated that Aurora-A regulates glycolytic rates in p53-deficient cells.

To confirm p53 deficiency is associated with Aurora-A medicated glycolytic regulation, p53 dominant-negative mutant R273H was expressed in A549 cells[26]. Remarkably, both the activity of Aurora-A and the level of glycolysis were significantly increased by the expression of R273H mutant (Fig. 1g, h). The increase in glycolysis was largely abolished after MLN8237 treatment, suggesting Aurora-A activity was required for this upregulation of glycolysis (Fig. 1g, h). Additionally, Aurora-A activity and the glycolytic rate are significantly higher in p53-null (p53−/−) than in p53-wild-type (p53+/+) HCT-116 cells. When Aurora-A was inhibited, the glycolysis was markedly decreased in p53−/− cells (Supplementary Fig. 1i, j). Together, these data revealed that upregulated Aurora-A promotes glycolysis in p53-deficient cancer cells.

**Aurora-A directly binds and phosphorylates LDHB.** Next, we sought to explore how Aurora-A modulates glycolysis. None of the glycolytic enzymes were upregulated at the protein level (Supplementary Fig. 2a), suggesting that Aurora-A may regulate glycolysis by modifying the activities of glycolytic enzymes. To explore the potential targets of Aurora-A in glycolytic regulation, we performed a co-immuno-precipitation (Co-IP) assay followed by mass spectrometry analysis (Fig. 2a). Intriguingly, while many known Aurora-A substrates and binding partners including TPX2, Bora, Cep192 et al. were precipitated with Aurora-A, glycolytic enzymes including PKM, PGAM1, ENOA, G3P, and LDHB were also identified in the precipitant (Supplementary Table 1). Because LDH controls glycolytic rates by generating lactate and $NAD^+$, it is one of most regulated enzymes in glucose metabolism. Additionally, it has been established that upregulation of LDHA significantly contributes to increased glycolysis in various cancers, however, how LDHB is regulated is poorly understood. Therefore, we wondered whether Aurora-A regulates glucose metabolism by modulating LDHB.

The association between Aurora-A and LDHB was first validated by reciprocal Co-IP assays with endogenous proteins. In p53-deficient cells, including U251, DLD1, Hela, and 293 T cells, the expression level of Aurora-A was high and Aurora-A interacted with LDHB. Whereas, in A549 and RKO cells that have wild-type p53, the expression of Aurora-A was low and the interaction between Aurora-A and LDHB was not detected (Fig. 2b and Supplementary Fig. 2b, c). Interestingly, Co-IPs using FLAG-tagged Aurora-A revealed that Aurora-A preferably interacts with LDHB over LDHA (Fig. 2c). Furthermore, data from the in vitro pull-down assays confirmed that Aurora-A directly interacts with LDHB and the binding affinity between Aurora-A and LDHB is much higher than the affinity between Aurora-A and LDHA (Fig. 2d). To examine their direct interaction in single cells, a FRET (fluorescence resonance energy transfer) assay was performed with mRuby2-Aurora-A and Clover-LDHA/B (Supplementary Fig. 2d). Consistently, FRET signals were detected in the cytoplasm of cells transfected with Aurora-A and LDHB, but not in cells transfected with Aurora-A and LDHA (Supplementary Fig. 2e). Domain mapping data revealed that the LDHB-C-terminus (amino acid 162–334)

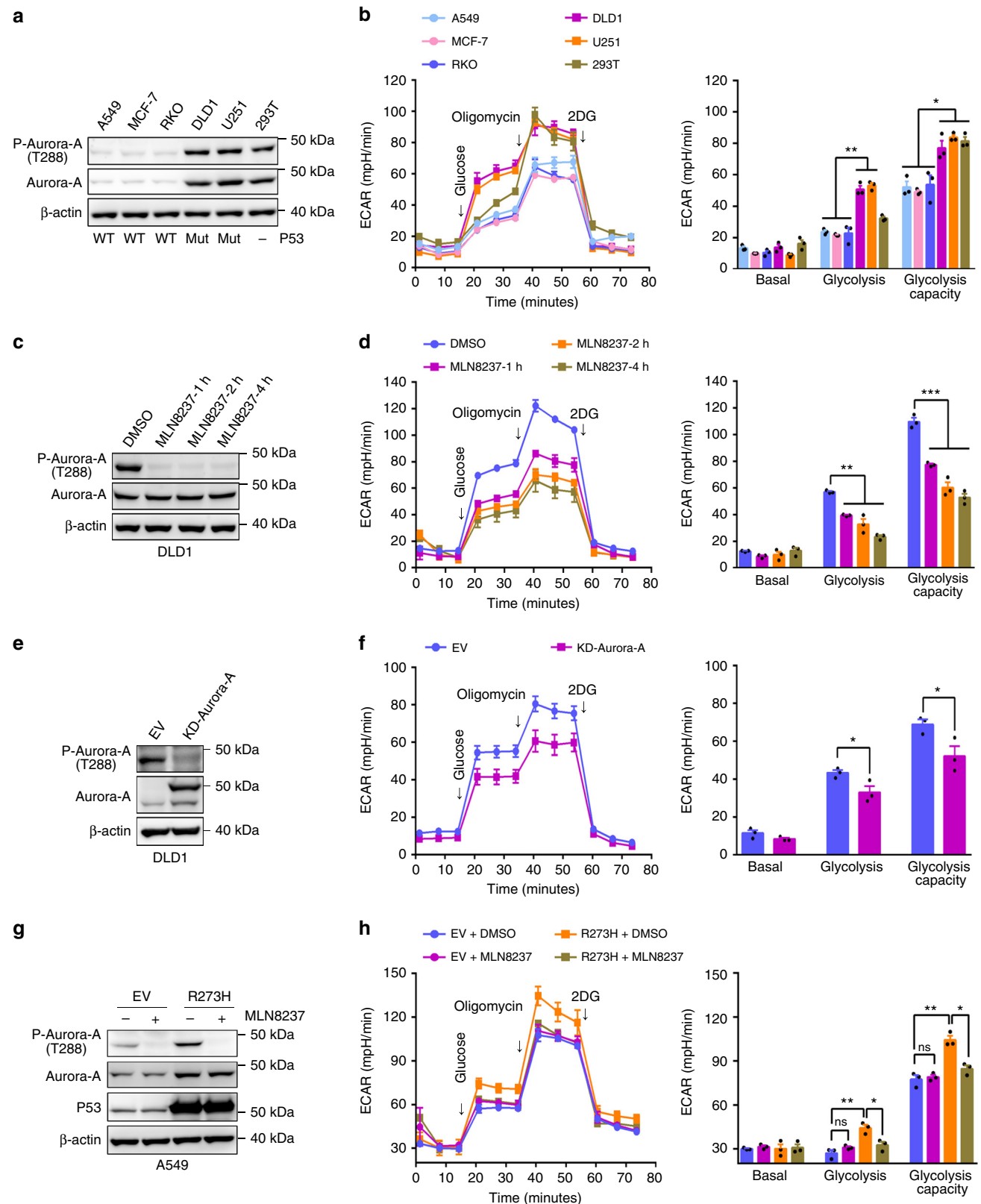

mediates the interaction with Aurora-A (Supplementary Fig. 2f, g). To further identify the interaction domain, various deletions in the LDHB C-terminus were constructed. When residues 199–250 within the substrate-binding domain of LDHB were removed, the interaction was abolished (Supplementary Fig. 2h). The corresponding region in LDHA contains three positively charged amino acids (Supplementary Fig. 2i, red asterisks) that are not present in LDHB, which may be responsible for their different affinities with Aurora-A. Notably, although the expression of LDHA is slightly higher than LDHB (LDHA/LDHB ranges from 1.2 to 1.4, except in U251 cells, in which LDHA is barely detected) in various cancer or transformed cells

**Fig. 1** High level of Aurora-A promotes glycolysis in malignant cancer cells. **a** In human cancer cell lines A549, MCF-7, RKO, DLD1, U251, and 293 T, the expressions and activity of Aurora-A kinase were examined. The status of p53 (WT: wild-type, Mut: mutation, -: inactivation) was labelled at the bottom of each lane. **b** The glycolytic rates were analyzed by standard Seahorse assay. The extracellular acidification rate (ECAR) over time (left panel) and ECAR in different stages of the measurement (right panel) were shown. **c** Aurora-A kinase activity was inhibited by selective inhibitor MLN8237 (200 nM, for 1, 2, and 4 h) in DLD1 cells. The kinase activity of Aurora-A was tested. **d** The glycolytic flux was investigated by seahorse assay in control and Aurora-A inhibition cells in **c**. The ECAR over time (left panel) and ECAR in different stages of the measurement (right panel) were shown. **e** Empty Vector (EV) and kinase-dead (KD, D274A) Aurora-A were transfected in DLD1 cells, then the levels of Aurora-A and p-Aurora-A were examined. **f** The ECAR of cells in **e** were measured by seahorse assay. **g** Empty Vector (EV) and p53 R273H mutant were transfected in A549 cells, the levels of Aurora-A, p-Aurora-A and p53 were examined. **h** The ECAR of cells in **g** were measured by seahorse assay. Cells were treated with DMSO or MLN8237 for 4 h before assay. The error bar in panels **b**, **d**, **f**, **h** represents the standard error of mean (SEM), $n = 3$ independent experiments. Source data are provided as a Source Data files. (Student's $t$-test *$p < 0.05$, **$p < 0.01$, ***$p < 0.001$, ns not significant).

(Supplementary Fig. 2j), LDHB, not LDHA, directly interacts with Aurora-A in the cytoplasm of p53-deficient cells.

Because the expression and the kinase activity of Aurora-A vary throughout the cell cycle with peaks during mitosis, we wondered whether its association with LDHB also shows cell cycle-dependent regulation. Unexpectedly, no FRET signal was detected in mitotic cells that expressed Clover-LDHB and mRuby-Aurora-A (Supplementary Fig. 2e). Co-IP was then performed in cells that were synchronized at various cell cycle stages (Supplementary Fig. 2k). Interestingly, the strongest association was detected in cells at S phase, when Aurora-A displayed modest activity (Fig. 2e). While Aurora-A activity increased in G2 phase, its association with LDHB decreased to a modest level. The associations are much weaker in G1 and M phases. As glycolysis and lactate production are enhanced in hypoxic condition, we wonder whether the association of Aurora-A and LDHB is increased in hypoxia. Indeed, when cells were cultured in hypoxic environment (Fig. 2f) or hypoxia induced factor 1 alpha (HIF1α) expression was induced by $CoCl_2$ treatment (Supplementary Fig. 2l), the association was enhanced without changes in the protein levels, indicating the interaction positively correlates with the glycolytic levels.

**Aurora-A phosphorylates LDHB at serine 162.** Inhibition of Aurora-A by MLN8237 or expression of KD-Aurora-A weakened its interaction with LDHB (Fig. 2g), indicating that the kinase activity of Aurora-A is required for a robust interaction. Therefore, we next tested whether LDHB is a direct substrate of Aurora-A. First, the in vitro kinase assay demonstrated that his-tagged LDHB was strongly phosphorylated by purified recombinant Aurora-A (Fig. 2h). To identify the phosphorylation sites, LDHB purified after the in vitro kinase assay were subjected to mass spectrometry analysis. Seven sites were identified to be phosphorylated by Aurora-A (Supplementary Table 2). These potential phosphorylation sites were then confirmed by MS analysis with the immuno-precipitated endogenous LDHB from various cells, including DLD1 and Hela (Supplementary Table 2). Remarkably, among the seven phosphorylation sites identified in vitro, only S162 (an evolutionarily conserved residue, Fig. 2i) were validated though it isn't a consensus site for Aurora kinases (Fig. 2j). Furthermore, the relative abundance of S162 phospho-peptide was significantly reduced when Aurora-A was knocked down in DLD1 and Hela cells (Fig. 2k and Supplementary Fig. 2m), indicating that Aurora-A phosphorylates LDHB at S162. Additionally, MS analysis confirmed that the relative abundance of S162 phosphopeptide is significantly higher in S phase than in M phase of the cell cycle (Supplementary Fig. 2n and Supplementary Table 2), further supporting that LDHB S162 is primarily phosphorylated in S phase. The phosphorylation of LDHB was also evaluated by pan-serine phosphorylation antibody. When Aurora-A was overexpressed, the serine-phosphorylation level was increased in wild-type LDHB but not

S162A-mutant-expressing cells (Fig. 2l). Together, these data demonstrate that Aurora-A phosphorylates LDHB at serine 162.

**S162 phosphorylation alters the enzymatic activities of LDHB.** It is known that LDHA prefers to catalyze the reduction of pyruvate to lactate (forward reaction), whereas LDHB has higher activity in the oxidation of lactate to pyruvate (reverse reaction) (Fig. 3a). Next, we sought to examine whether S162 phosphorylation modulates the enzymatic activities of LDHB. Endogenous LDHB was first depleted by shRNA and replaced by shRNA-resistant and FLAG-tagged LDHB-WT or LDHB-S162A/D mutants (Fig. 3b), then FLAG-tagged LDH was purified by immune-precipitation for enzymatic activity measurements (Supplementary Fig. 3a). Indeed, LDHA showed notably higher activity in the forward reaction and lower activity in the reverse reaction when compared with LDHB (Fig. 3c). Surprisingly, while the activity of the non-phospho mutant LDHB-S162A showed moderate changes in the two directions, the activity of the phospho-mimic mutant LDHB-S162D exhibited an approximate three-fold increase in the forward reaction (higher than that of LDHA) and an ~50% decrease in the reverse reaction (Fig. 3c). Furthermore, the recombinant LDHB-S162D expressed in *E. coli* showed similar trends but greater changes for both reactions (Fig. 3d). Consistently, overexpressing Aurora-A promoted the forward reaction and inhibited the reverse reaction of LDHB (Fig. 3e). By contrast, overexpressing KD-Aurora-A decreased the activity of the forward reaction, but increased the activity of the reverse reaction of LDHB (Fig. 3e). However, Aurora-A overexpression did not alter the activities of LDHB-S162A in the two directions, demonstrating that Aurora-A modulates LDHB by phosphorylating serine 162, rather than other sites (Fig. 3e). In line with these data, after pre-incubation with Aurora-A (allosteric activated by TPX2 1-25 amino acid[20], Supplementary Fig. 3b) and ATP in vitro, LDHB showed opposite alterations in two reactions: increase in the forward reaction and decrease in the reverse reaction (Fig. 3f). The activity of LDHB-S162D (Supplementary Fig. 3c) and LDHB with overexpressed Aurora-A (Supplementary Fig. 3d) were also examined in U251 cells, whose LDH comprises the LDHB subunit exclusively (Supplementary Fig. 2j). Similar trends but greater levels of alterations in the two directions of the activities of LDHB were observed (Supplementary Fig. 3c, d). In line with the interaction data (Fig. 2f and Supplementary Fig. 2l), HIF1α stabilization led to increased activity of forward reaction and decreased activity in reverse reaction (Supplementary Fig. 3e).

To examine the enzymatic activities in live cells, a redox biosensor was applied to monitor the intracellular ratio of NADH/NAD$^+$ in real time[27]. Shortly after the addition of 5 mM pyruvate into the medium of LDHB-WT expressing cells, the ratio of NADH/NAD$^+$ (represented as F425/485) rapidly declined with the reduction of pyruvate to lactate by LDH (Fig. 3g and blue line in 3 h). However, this rapid change in

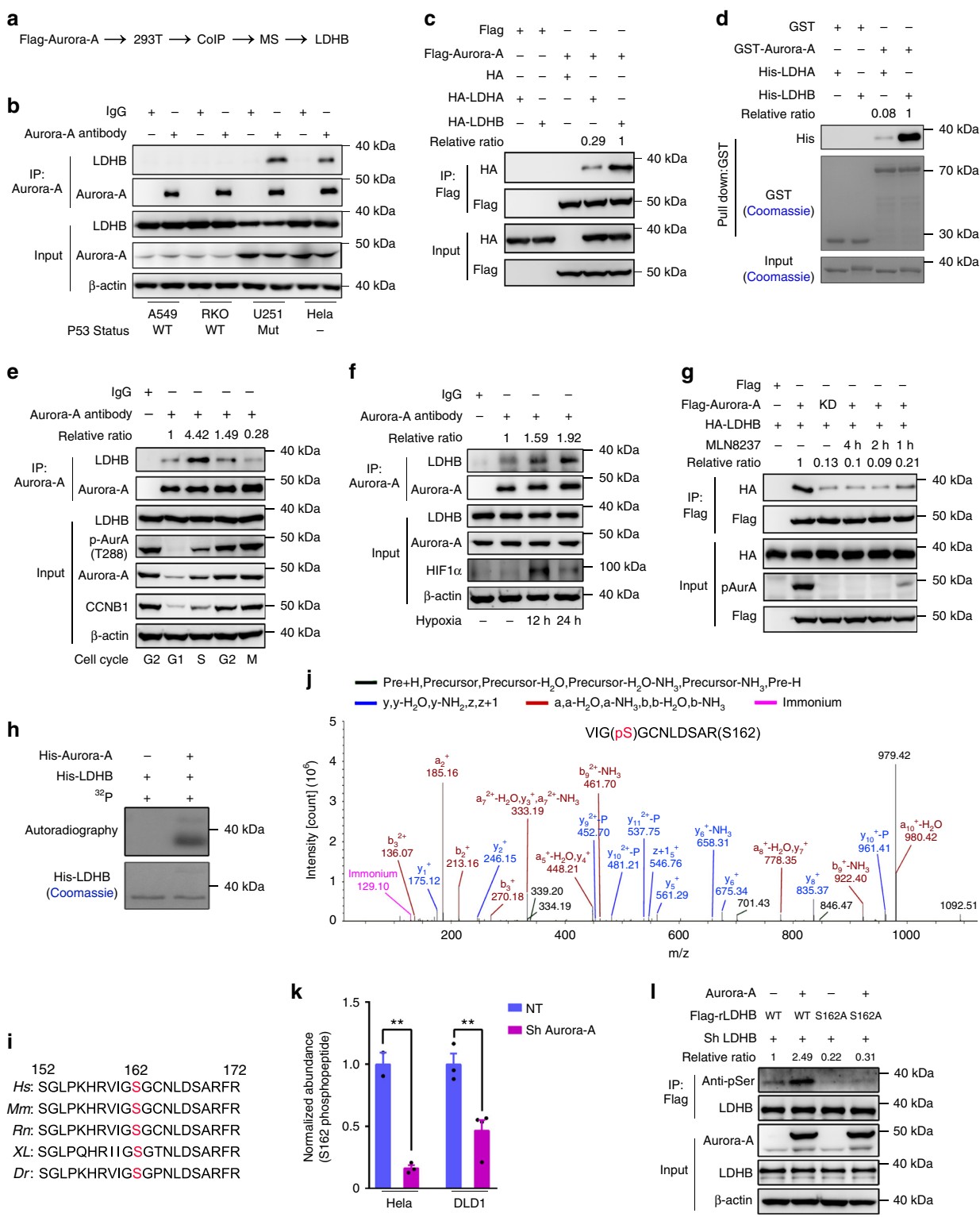

NADH/NAD$^+$ was markedly repressed after Aurora-A inhibition by MLN8237 for 1, 2 (Supplementary Fig. 3f), or 6 h (Fig. 3g and purple line in 3 h), indicating that the rates of the forward reaction were markedly decreased, which confirmed that Aurora-A modulated LDH through phosphorylation. Consistently, enhancing LDHB S162 phosphorylation via overexpression of Aurora-A promoted the oxidation of NADH to NAD$^+$ (Fig. 3i, j). In agreement with the activities analysis data (Fig. 3c–e), the decline in NADH/NAD$^+$ was faster in LDHB-S162D expressing cells (Fig. 3g and orange line in 3 h), suggesting its higher activity

in the forward reaction. Importantly, this acceleration effect was not affected by Aurora-A inhibition (Fig. 3g and brown line in 3 h), further demonstrating that phosphorylation of S162, rather than other sites, is mainly responsible for the increased activity of the forward reaction (Fig. 3g, h). When LDHA was knocked down in LDHB-S162D expressing cells, the activity of forward reaction was not significantly compromised, suggesting LDHB-S162D is largely responsible for the acceleration of the forward reaction (Supplementary Fig. 3g). Thus, these data from live cells confirmed that Aurora-A mediated phosphorylation of

**Fig. 2** Aurora-A directly binds and phosphorylates LDHB at serine 162. **a** Schematic diagram of the procedure to seek for Aurora-A associated proteins in glycolytic regulation. Identified proteins were listed in Supplementary Table 1. **b** Co-immune-precipitation (Co-IP) was conducted in A549, RKO, U251 and Hela cells. The status of p53 was indicated at the bottom of each lane. **c** FLAG-tagged Aurora-A was co-expressed with HA-tagged LDHA/B in 293 T cells. Co-IP was conducted with FLAG-beads followed by WB. **d** GST pull-down assay was performed with GST-Aurora-A and His-LDHA/B, followed by Coomassie blue staining and WB. **e** HeLa cells were synchronized at different cell cycle stages. Co-IP was conducted with Aurora-A antibody followed by WB. Relative ratio means the binding affinity between LDHB and Aurora-A. **f** DLD1 was cultured in hypoxic incubator for 12 and 24 h. Co-IP was conducted with Aurora-A antibody. **g** FLAG-tagged Aurora-A or KD-Aurora-A were co-expressed with HA-tagged LDHB in 293 T cells. Aurora-A was inhibited by MLN8237. Co-IP was conducted with FLAG-beads. **h** Recombinant His-Aurora-A and His-LDHB were incubated with $^{32}$p labeled ATP followed by SDS-PAGE. The gel was subjected to autoradiography and Coomassie blue staining. **i** Alignment of amino acid sequences containing serine 162 of LDHB from several model species. *Dr (Danio rerio); XL (Xenopus laevis); Rn (Rattus norvegicus); Mm (Mus musculus); Hs (Homo Sapiens)*. **j** Mass spectrometry data showed serine 162 of LDHB was phosphorylated in cultured tumor cells. A tryptic fragment at $m/z$ 636.26813 ($z = +2$), matched to the charged peptide VIGS(ph)GCNLDSAR. **k** Aurora-A was knocked down in DLD1 and Hela cells. The endogenous LDHB was isolated and subjected for MS analyses. The relative abundance of S162 phosphopeptide was quantified. **l** Endogenous LDHB was knocked down in DLD1 cells, and shRNA-resistant FLAG-LDHB WT, S162A mutant and Aurora-A were expressed as indicated. LDHB was purified with FLAG beads. The error bar in panels **k** represents the standard error of mean (SEM), $n = 2$ independent experiments for Hela NT (non-targeting), $n = 3$ for Hela Sh Aurora-A and DLD1 NT and $n = 4$ for DLD1 sh Aurora-A. Source data are provided as a Source Data files. (Student's $t$-test $**p < 0.01$).

LDHB S162 markedly enhances its catalytic activity of the forward reaction.

To explore how the activities of LDHB were changed upon Aurora-A-mediated phosphorylation, we first examined the isozyme composition of tetrameric LDH. Native gel electrophoresis of LDH isozymes showed that the majority of LDH in DLD1 cells is LDH3, containing two LDHA and two LDHB subunits (Supplementary Fig. 3h). While LDHA/B knockdown shifted the isozyme proportions, neither the expression of WT nor KD Aurora-A altered the composition of the isozymes (Supplementary Fig. 3h). This result was confirmed with Co-IP assays in which the associations of exogenous LDHB with endogenous LDHA/B were examined. The ratios of endogenous LDHA/B to FLAG-LDHB in the precipitant did not change after Aurora-A overexpression (Supplementary Fig. 3i). Additionally, while LDHA or LDHB depletion markedly altered the ratios of endogenous LDHA/B to FLAG-LDHB, these ratios remained unchanged when LDHB wild-type or S162A/D mutants were expressed (Supplementary Fig. 3j), indicating that the composition of the LDH isozyme was not regulated by Aurora-A-mediated phosphorylation.

**Phosphorylation of LDHB S162 relieves substrate inhibition.** To dissect the mechanism underlying the alterations in LDHB activity after S162 phosphorylation, the enzymatic kinetics were evaluated. First, the binding affinities ($K_d$) of the wild-type LDHB and S162D mutant towards NADH and oxamate (pyruvate analogue) were assessed by ITC (isothermal titration calorimetry) assay. As shown in Fig. 4a, b, the S162D mutant has larger (approximately five-fold) $K_d$ values towards both NADH and oxamate than wild-type LDHB, indicating that phosphorylation of S162 somewhat reduces the affinities of LDHB with NADH and pyruvate. To seek structural insights into the affinity between LDHB and NADH upon Ser162 phosphorylation, dynamic conformation was simulated using the structure of LDHB (PDB code: 1I0Z). Molecular modeling revealed that phosphorylation of LDHB at Ser162 pushes the nicotinamide ring of NADH away from the activity center (Supplementary Fig. 4a, b), suggesting a lower affinity of LDHB with NADH, which is in line with the ITC data (Fig. 4a). The simulation result also suggested that the frequency of a hydrogen bond between LDHB pS162 and nicotinamide ring of NADH was significantly reduced in the modeling period of 50 nanosecond (Supplementary Fig. 4c). Hence, these results suggested that the activity of forward reaction might be compromised when LDHB S162 is phosphorylated, which is inconsistent with the measurement data (Fig. 3 and Supplementary Fig. 3).

Interestingly, the activity of LDHB has been reported to be inhibited by excess pyruvate, the so-called substrate-inhibition effect[28–31]. This inhibition could be largely removed if the serine 162 is replaced by leucine[32,33], which prompted us to explore whether serine 162 phosphorylation modulates the substrate-inhibition effect. Therefore, the correlation between LDHB activity and the pyruvate concentration was analyzed. Indeed, the dose curves of the pyruvate concentration for LDHA and LDHB activities are bi-phasic. Enzymatic activity increases with the pyruvate concentration to specific points, after which the activity decreases. The maximal activities were shown at different pyruvate concentrations for LDHB (~0.25 mM) and LDHA (~1.25 mM), consistent with previous report that LDHB has stronger substrate-inhibition effect[34,35]. Surprisingly, LDHB-S162D showed highest activity when the pyruvate concentration is 10 mM, which is much higher than those for LDHB and LDHA, suggesting that when the pyruvate concentration is between 0.25 and 10 mM, the substrate-inhibition is mostly relieved after LDHB S162 phosphorylation. Notably, in DLD1 cells, the intracellular concentration of pyruvate was ~0.7 mM, and was increased to ~1.9 mM after treated with $CoCl_2$ or to ~1.2 mM when the glycolysis was promoted by Oligomycin (Supplementary Fig. 4d). Therefore, within the physiological range of intracellular pyruvate concentration (0.25–5 mM)[36,37], while LDHB activity in reducing pyruvate to lactate was inhibited by pyruvate, the activity of the LDHB-S162D mutant was significantly increased (Fig. 4c), suggesting that S162 phosphorylation relieves the substrate-inhibition and promotes the activity of the forward reaction. Next, the kinetic feature of LDHB was evaluated by plotting the catalytic rate of the forward reaction over physiological concentration of pyruvate (Fig. 4d). The kinetic curves revealed that S162D mutant works more like LDHA, which is consistent with the alterations of the enzymatic activity (Fig. 3c–f). Thus, when LDHB S162 was phosphorylated by Aurora-A in the presence of high level of pyruvate, the reduction of pyruvate to lactate was markedly accelerated (Fig. 4c). Together, these results revealed that LDHB S162 phosphorylation releases the substrate-inhibition by pyruvate, therefore boosting the catalytic efficiency of lactate generation in cancer cells.

**LDHB phosphorylation promotes glycolysis and biosynthesis.** Next, we wondered the function of LDHB S162 phosphorylation in metabolic regulation. Since LDHB S162 phosphorylation promotes the reduction of pyruvate to lactate, it is expected that the glycolytic rate could be enhanced when LDHB is phosphorylated by Aurora-A. Indeed, as in the LDHB depletion cells

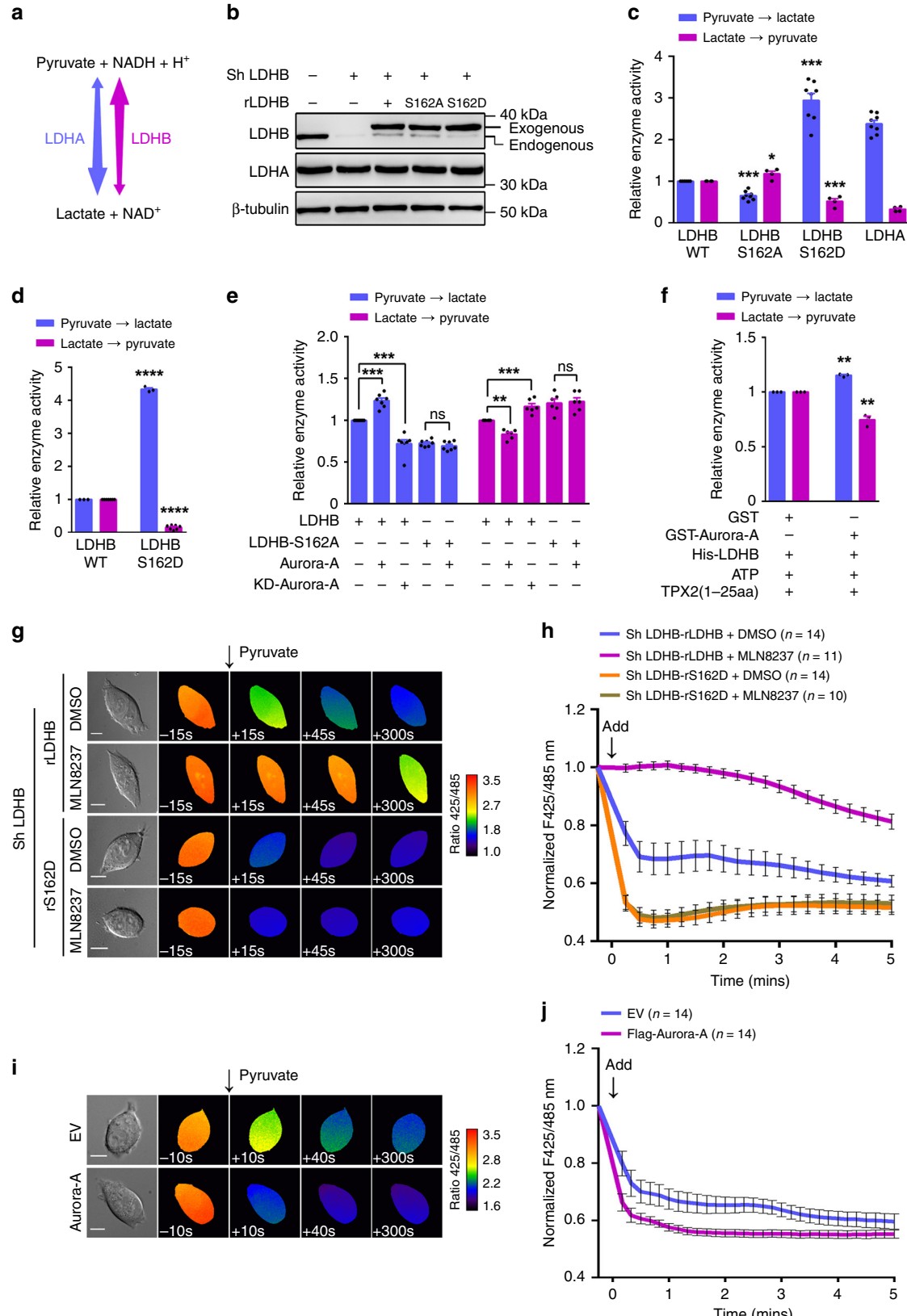

(Fig. 5a, b), the non-phosphorylated mutant LDHB-S162A expressing cells showed a decreased glycolytic rate (Fig. 5c, d). This decrease could be ascribed to the reduction in LDH activity of forward reaction because additional expression of LDHA fully rescued the glycolytic rate (Fig. 5c, d). In contrast to S162A,

S162D increased the glycolytic flux (Fig. 5c, d). Moreover, Aurora-A inhibition reduced the glycolytic flux in LDHB-WT-expressing cells, but not in LDHB-S162D-expressing cells, confirming that phosphorylation of S162 was required for Aurora-A mediated regulation of LDHB in glycolysis (Fig. 5e, f).

**Fig. 3** Phosphorylation of LDHB S162 alters its enzymatic activities. **a** Diagram of the bi-directional reactions catalyzed by tetrameric LDH, comprising LDHA and LDHB. **b** In DLD1 cells, the endogenous LDHB was replaced by shRNA-resistant and FLAG-tagged LDHB WT or S162A/D mutants. The expressions of LDHA/B were examined by WB. **c** FLAG-tagged WT, S162A, S162D of LDHB and LDHA were purified by IP and subjected to measure the bi-directional activities. **d** His-tagged LDHB WT and S162A were expressed in *E. coli*. The purified proteins were subjected to measure the bi-directional activities. **e** WT-Aurora-A or KD-Aurora-A was expressed in the DLD1 cells tested in **b**, FLAG-tagged proteins were purified by IP and subjected to measure the bi-directional activities. **f** His-tagged LDHB was incubated with ATP, TPX2(1–25aa) and GST or GST-Aurora-A. Bi-directional activities of LDHB were measured. **g** The plasmid of NADH/NAD$^+$ sensor SoNar was transfected into DLD1 cells used in **b**. Cells were subjected to live cell imaging. Two channels of emission signals were collected near-simultaneously at 15 s intervals before and after addition of pyruvate (5 mM). The differential interference contrast (DIC) images of the cells (left) and the ratio images of F425/485 were shown. Aurora-A was inhibited by MLN8237 (200 nM, 6 h) before imaging. Scale bar, 10 μm. **h** Quantitation of NADH/NAD$^+$ ratio (presented as F425/485) in **g**. Arrow indicates the addition of pyruvate at time 0. **i** NADH/NAD$^+$ ratios were measured in empty vector (EV) and Aurora-A overexpressing cells at 10 s intervals. DIC and ratio images of F425/485 were shown. Scale bar, 10 μm. **j** Quantitation of NADH/NAD$^+$ ratio in **i**. The error bar in panels **c**, **d**, **e**, **f** represents the standard error of mean(SEM), $n = 4$ independent experiments in panels **c**, 3 in panels **d**, **f** and 7 in panels **e**. Source data are provided as a Source Data files. (Student's *t*-test *$p < 0.05$, **$p < 0.01$, ***$p < 0.001$, ns not significant).

Furthermore, LDHB-WT and LDHB-S162D, but not LDHB-S162A, restored the decrease of glucose uptake and lactate production when endogenous LDHB was knocked down (Supplementary Fig. 5a). Because sustained NAD$^+$ regeneration is essential for glycolysis flux[2], the NADH/NAD$^+$ ratio was analyzed with SoNar sensor. Indeed, LDHB-S162D expression facilitated NAD$^+$ regeneration, which is evidenced by the decrease in the NADH/NAD$^+$ ratio (Supplementary Fig. 5b). By contrast, LDHB-S162A increased NADH/NAD$^+$ ratio, and this phenotype was rescued by co-expression of LDHA, suggesting NAD$^+$ regeneration was compromised by LDHB-S162A (Supplementary Fig. 5b). Notably, the oxygen consumption rate was not affected in LDHB-S162D expressing cells or in cells treated with MLN8237 (Supplementary Fig. 5c). Together, these data demonstrated that Aurora-A mediated LDHB S162 phosphorylation promotes glycolysis.

In cancer cells, glycolytic intermediates provide building-blocks for biosynthesis pathways such as the pentose phosphate pathway (PPP)[4,38]. Therefore, we tested whether intermediate metabolites and products in the biosynthesis pathways were changed accordingly by tracing $^{13}$C-labeled glucose. Consistently, the proportions of $^{13}$C incorporated glucose-6-phosphate (G6P), phosphoenolpyruvate (PEP), lactate and ribose-5-phosphate (R5P) were decreased in LDHB-S162A expressing cells, and this decrease was rescued by simultaneously overexpressing LDHA (Fig. 5g). By contrast, the proportions of $^{13}$C incorporated G6P, PEP, R5P, and lactate were increased in LDHB-S162D expressing cells (Fig. 5h). Importantly, Aurora-A inhibition significantly lowered the fraction of $^{13}$C-glucose-derived carbon into G6P, PEP, R5P, and lactate in LDHB-WT-expressing but not in LDHB-S162D mutant-expressing cells (Fig. 5h), confirming that Aurora-A modulates glycolytic flux by phosphorylating LDHB S162. Moreover, while LDHB-S162D expression increased the levels of G6P, 2PG, NADPH and GSH, LDHB-S162A expression reduced the levels of these metabolites and this reduction could be rescued by additionally expressed LDHA (Supplementary Fig. 5d). Consistently, Aurora-A inhibition notably reduced NADPH and GSH levels (Supplementary Fig. 5e).

Together, these results demonstrated that Aurora-A-mediated LDHB S162 phosphorylation promotes NAD$^+$ regeneration, glycolytic flux, lactate production and biosynthesis with glycolytic metabolites in cancer cells.

**LDHB S162 phosphorylation is required for tumor progression.** To evaluate the functional significance of S162 phosphorylation of LDHB, the proliferation of cells expressing LDHB-S162A/D was tested in cultured cells and xenograft tumors. As expected, the growth of DLD1 cells was remarkably suppressed after LDHB depletion (Fig. 6a, b). While reintroduction of RNAi-

resistant LDHB-WT and LDHB-S162D fully restored cell growth, LDHB-S162A expression failed (Fig. 6a, b), suggesting that LDHB S162 phosphorylation is required for the rapid proliferation of tumor cells. Cell cycle analysis showed that the proportion of cells at S phase markedly increased upon LDHB-S162A expression, suggesting S162 phosphorylation is important for S phase progression (Supplementary Fig. 6a, b). Results of colony formation assay indicated that LDHB-S162D, not LDHB-S162A, promoted cell growth. Additional LDHA expression rescued the colony formation ability of LDHB-S162A, suggesting its defects in reduction of pyruvate to lactate (Fig. 6c). Next, we determined the role of S162 modification in xenograft tumors. Consistently, progression of LDHB-S162A-expressing tumor was notably diminished (Fig. 6d–f) and this inhibition was mostly recovered by excess LDHA expression (Fig. 6d–f), indicating that S162 phosphorylation of LDHB promotes tumor progression by upregulating glycolytic flux, lactate production and NAD$^+$ regeneration, the same mechanism by which upregulated LDHA promotes aerobic glycolysis and tumor progression. Similar results were obtained with U251 (Supplementary Fig. 6c–g) and HeLa cells (Supplementary Fig. 6h–k). To determine the proliferation capability of the tumor, Ki67 levels were examined in tumors sections. Consistently, the indexes of Ki67-positive cells were proportional to the weight of the xenograft tumors (Fig. 6g, h).

**Correlation of Aurora-A and LDHB in clinical samples.** To further determine the clinical relevance of Aurora-A-LDHB regulation, we analyzed the expression of Aurora-A and LDHB from published GEO databases. Data from multiple datasets showed that both LDHB and Aurora A are significantly upregulated in various tumors (Supplementary Fig. 6l–n). Remarkably, the expression of Aurora-A and LDHB showed a significant positive correlation in colon cancer samples with p53 mutation, but not in normal samples (Supplementary Fig. 6l). Similar results were found in lung and cervical cancer samples (Supplementary Fig. 6m, n). Furthermore, immuno-histochemical staining of consecutive tumor sections showed that Aurora-A and LDHB are simultaneously overexpressed in colon cancers (Supplementary Fig. 6o). Together, these data indicate that Aurora-A and LDHB overexpression are correlated in human patients.

Together, we demonstrated that Aurora-A phosphorylates LDHB at Ser162, facilitating glycolysis and biosynthesis to promote tumor growth. Moreover, in clinical samples, the expression of Aurora-A and LDHB is strongly correlated. Thus, in normal tissue the expression of Aurora-A is relatively low, and the glycolytic level is at the basal level (Fig. 6i). During malignant progression, upregulated Aurora-A phosphorylates LDHB, which releases its substrate-inhibition by pyruvate and

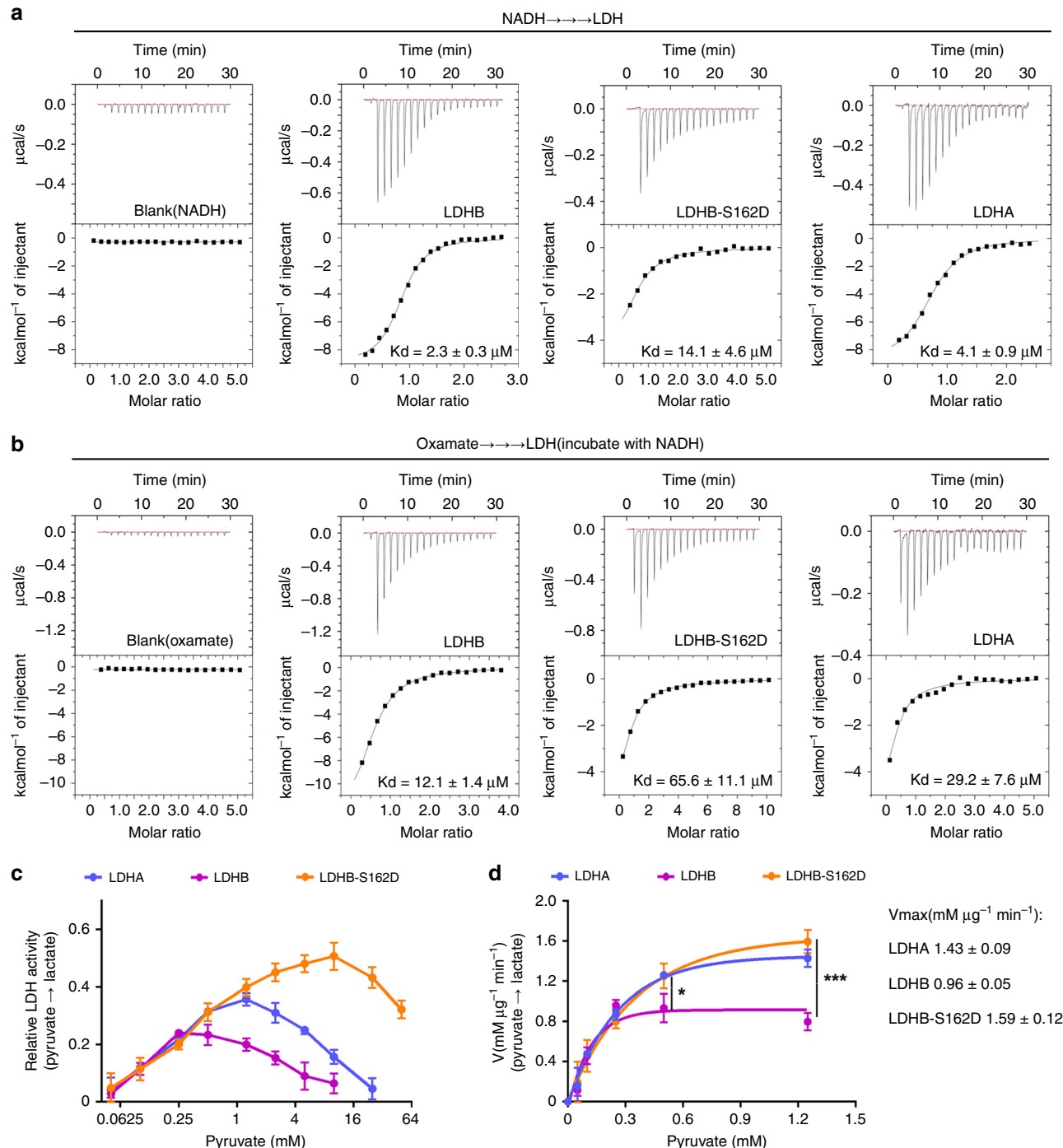

**Fig. 4** The substrate-inhibition effect was relieved by LDHB-S162D. **a** The dissociation constant of NADH for LDH was assessed by isothermal titration calorimetry (ITC) assay. ITC results were shown for interactions of NADH with LDHB, LDHB-S162D, and LDHA. The $K_d$ values were labeled at the bottom. **b** ITC results were shown for oxamate interactions with LDHB-NADH complex, LDHB-S162D-NADH complex and LDHA-NADH complex. The $K_d$ values were labeled at the bottom. **c** The catalytic activity (Pyruvate to lactate) of LDHA, LDHB, and LDHB S162D were measured at different concentrations of pyruvate. The relative enzyme activity for LDHB, LDHB-S162D, and LDHA were plotted against the concentrations of pyruvate (logarithm of 2). **d** The catalytic rates of the forward reactions for LDHA, LDHB and LDHB S162D were plotted over physiological concentrations of pyruvate. Student $t$-test were performed between the rate of LDHB and LDHB S162D at 0.5 and 1.25 mM pyruvate. Curve fitting was conducted using Prism software. The error bar in panels **a**, **b**, **c**, **d** represents the standard error of mean(SEM), $n = 6$ independent experiments for LDHB, 7 for LDHB-S162D, 4 for LDHA in panels **a**, $n = 4$ independent experiments for LDHB, 6 for LDHB-S162D, 4 for LDHA in panels **b**, $n = 4$ independent experiments for LDHB, 3 for LDHB-S162D, 4 for LDHA in panels **c** and **d**. Source data are provided as a Source Data files. (Student's $t$-test *$p < 0.05$, **$p < 0.01$, ***$p < 0.001$, ns not significant).

promotes the reduction of pyruvate to lactate. This enzymatic change of LDHB leads to significant increase in NAD$^+$ regeneration, glycolysis flux and lactate production. Meanwhile, more glycolytic intermediates are used for biosynthesis (Fig. 6i).

These alterations facilitate rapid tumor progression in multiple cancers. Thus, this is a mechanism by which the Warburg effect is efficiently modulated by Aurora-A kinase-mediated LDHB phosphorylation.

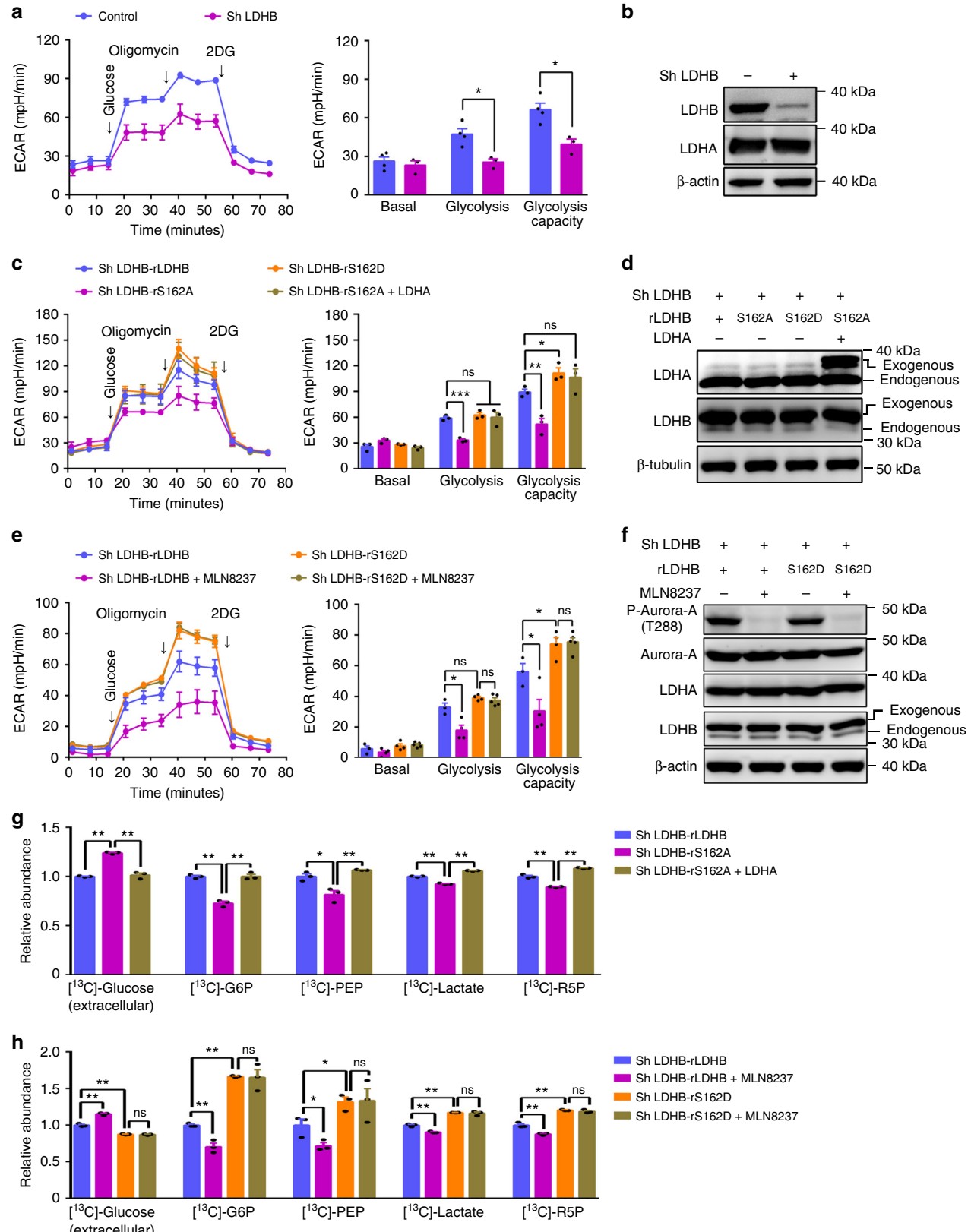

## Discussion

In this study, we uncovered a mechanism by which glycolysis and biosynthesis are efficiently promoted by Aurora-A-mediated LDHB phosphorylation. This regulation mainly occurs in p53-deficient cells, in which Aurora-A is upregulated after relieved from p53 mediated inhibition[23]. In p53 competent cells, it directly suppresses the transcription of Aurora-A[39] or indirectly through Rb-E2F3 pathway[40]. p53 also inhibits the activity of Aurora-A via Gaddd45a[41], and promotes the degradation of Aurora-A via FBXW7[42]. Therefore, Aurora-A is significantly upregulated in p53-deficient cells. Because p53 inhibits glycolysis via multiple pathways[43], glycolysis is promoted when p53 is

**Fig. 5** Phosphorylation of LDHB S162 promotes glycolysis. **a** In DLD1 cells, LDHB was knocked down by sh RNA. Seahorse assays were performed to evaluate the glycolytic conditions. ECAR over time (left panel) and ECAR in different stages of the measurement (right panel) were shown. **b** The expression of LDHA and LDHB in cells used in **a** were examined by WB. **c** In DLD1 cells, endogenous LDHB was knocked down, then shRNA-resistant WT LDHB, LDHB S162A/D, and LDHA were expressed. Seahorse assays were performed in these cells to evaluate the glycolytic flux. ECAR over time (left panel) and ECAR in different stages of the measurement (right panel) were shown. **d** The expression of LDHA/B in cells used in **c** were examined by WB. **e** In DLD1 cells used in **c**, Aurora-A was inhibited by MLN8237 (200 nM, 4 h). Seahorse assays were conducted. ECAR over time (left panel) and ECAR in different stages of the measurement (right panel) were shown. **f** The expression of LDHA/B and the activity of Aurora-A kinase in cells used in **e** were examined by WB. **g** The glycolytic tracing assay was performed with [13]C-labeled glucose in cells used in **c**. The relative abundance of the [13]C-labeled glycolytic metabolites: Glucose-6-phosphate (G6P), phosphoenolpyruvate (PEP), lactate and ribose-5-phosphate (R5P) were shown. **h** The glycolytic flux assay was performed with [13]C-labeled glucose in cells used in **e**. MLN8237 (200 nM, 24 h) was used to inhibit Aurora-A activity. The relative abundance of the [13]C-labeled glycolytic metabolites were shown. The error bar in panels **a**, **c**, **e**, **g**, **h** represents the standard deviation (SD), $n = 3$ biologically independent samples. Source data are provided as a Source Data files. (Student's t-test *$p < 0.05$, **$p < 0.01$, ***$p < 0.001$, ns not significant).

inactivated. Furthermore, as p53 promotes pyruvate utilization in mitochondria[44], the concentration of pyruvate is increased in p53-deficient cells, which could enhance the substrate-inhibition of LDH. Therefore, when p53 is deficient, upregulated Aurora-A phosphorylates LDHB to relief the substrate-inhibition, leading to increases in lactate production and NAD$^+$ regeneration. Consequently, glycolytic flux and biosynthesis are enhanced, which facilitates tumor cell proliferation.

LDH has been found to be strictly controlled in cancer cells. The expression of LDHA subunit was reported to be upregulated by oncogene such as c-Myc and HIF1[9,45]. The activity of LDHA is also promoted by modification, such as phosphorylation at tyrosin10[7,46,47] and deacetylation at lysine5[8]. By contrast, few studies have reported that LDHB is inhibited by promoter methylation[48,49]. Thus, it is thought that glycolysis is promoted by upregulation of LDHA and downregulation of LDHB. Although this model explains the metabolic phenotypes in several cancers, such as liver, brain, and pancreatic cancer[50–52], it is not supported by clinical data from colon, breast, and lung cancers, in which LDHB is also abundant[10,11,53]. Meanwhile, increasing evidence has shown that LDHB is critical for malignant progressions in triple-negative breast cancer[11], K-Ras amplified lung cancer[10] and colon cancer[53]. However, how LDHB functions in these cancers is poorly understood. Intriguingly, our findings provide a mechanism by which LDHB modification by Aurora-A releases its substrate-inhibition and directly contributes to the Warburg effect.

Although the expression of LDHB is slightly less than LDHA in DLD1 and HeLa cells, LDHB is indispensable for high level of glycolysis (Fig. 5a and Supplementary Fig. 5a) and tumor progression (Fig. 6d–f and Supplementary Fig. 6c-k). Despite the protein is abundant, LDHA is not sufficient to support the increased demand of glycolysis. In line with previous reports[31,33,34], our data revealed LDHA exhibits the similar substrate-inhibition as LDHB does, though the inhibitory concentration of pyruvate is higher for LDHA (Fig. 4c). The catalytic activity of LDHA was gradually inhibited when the concentration of pyruvate was increased from 1.25 to 5 mM (Fig. 4c), within the physiological concentration[36,37]. Therefore, the compromised activity may not satisfy the requirement of high glycolytic demand. In addition, glycolysis is promoted in hypoxic microenvironment, which also promotes the substrate-inhibition of LDH. Aurora-A mediated LDHB phosphorylation significantly relieves the substrate inhibition, resulting in rapid NAD$^+$ regeneration and upregulation of glycolysis. Notably, Aurora-A phosphorylates LDHB in S phase, when biosynthesis building-block and NADPH are intensively demanded. Thus, in p53-deficient cells, upregulation of lactate production through LDHB phosphorylation could be a very efficient way to promote overall glycolysis, biosynthesis and tumor growth (Fig. 6i). Since LDHB was proved to promote autophagy by interacting with V-ATPase

at lysosome[13], we investigated whether LDHB-S162A compromised autophagy. No change in the level of LC3-II, the autophagy marker, was observed, in cells expressing LDHB-S162A/D mutants (Supplementary Fig. 6p, q), indicating that LDHB-S162 phosphorylation does not affect autophagy.

In the LDH tetramer, LDHB S162 is in the center of the catalytic domain and not easily accessible for Aurora-A kinase. According to the immune-precipitation data (Fig. 2c), the Aurora-A-LDHB complex contains little LDHA. So it is possible that Aurora-A interacts and phosphorylates LDHB when it is in a transient state of monomer. Moreover, the residue serine162 locates at a non-structure region (between α-helix and β-sheet), which might be more flexible and accessible for Aurora-A.

Remarkably, it was reported that replacement of LDHA/B S162 by leucine efficiently relieved the substrate-inhibition by pyruvate[32,33], indicating S162 is indeed critical for the substrate-inhibition of LDHB. It was suggested that the change from serine to leucine reduces the affinity between LDHB and NADH, and disrupts the formation of the LDHB-pyruvate-NAD$^+$ adduct[32,33], a covalent adduct causing the substrate-inhibition effect[28–30,35]. In our study, molecular modeling showed that the nicotinamide ring of NADH is pushed away from S162 of LDHB after the serine is phosphorylated (Supplementary Fig. 4a, b). Moreover, the ITC data indicated that the affinities between LDHB and substrates (NADH and oxamate) are reduced when LDHB S162 is mutated to aspartic acid (Fig. 4a), which is in line with the data that low affinities between substrates and LDH release the substrate-inhibition effect in early studies[32,33,35]. Therefore, phosphorylation of S162 might cause conformational changes in the complex of LDHB-NADH-pyruvate, which decrease the affinities of LDHB with both NADH and pyruvate. These alterations might disrupt the formation of LDHB-pyruvate-NAD$^+$ adduct and release the substrate-inhibition. How the conformational alterations release the substrate-inhibition need be explored by further work.

Two recent articles reported that circulating lactate is the primary carbon source for the TCA cycle in multiple tissues and cancers[54,55]. On the other hand, lactate production and NAD$^+$ regeneration are required to sustain glycolytic flux during tumor progression. Thus, switching the catalytic direction of LDHB would be an efficient strategy for cancer cells to fit fluctuating metabolic demands. In this regard, our current work provides such a mechanism by which the primary catalytic activity of LDHB is rapidly changed by oncogenic Aurora-A mediated phosphorylation, a reversible process that is faster than transcriptional regulation. Blocking this modification could potentially interrupt lactate metabolism in cancer cells, resulting in tumor inhibition. Aurora-A kinase and LDH inhibitors are currently undergoing clinical trials. The combination of inhibitors for Aurora-A and LDH could be a promising therapeutic strategy for glycolytic tumors with high Aurora-A and LDHB expression.

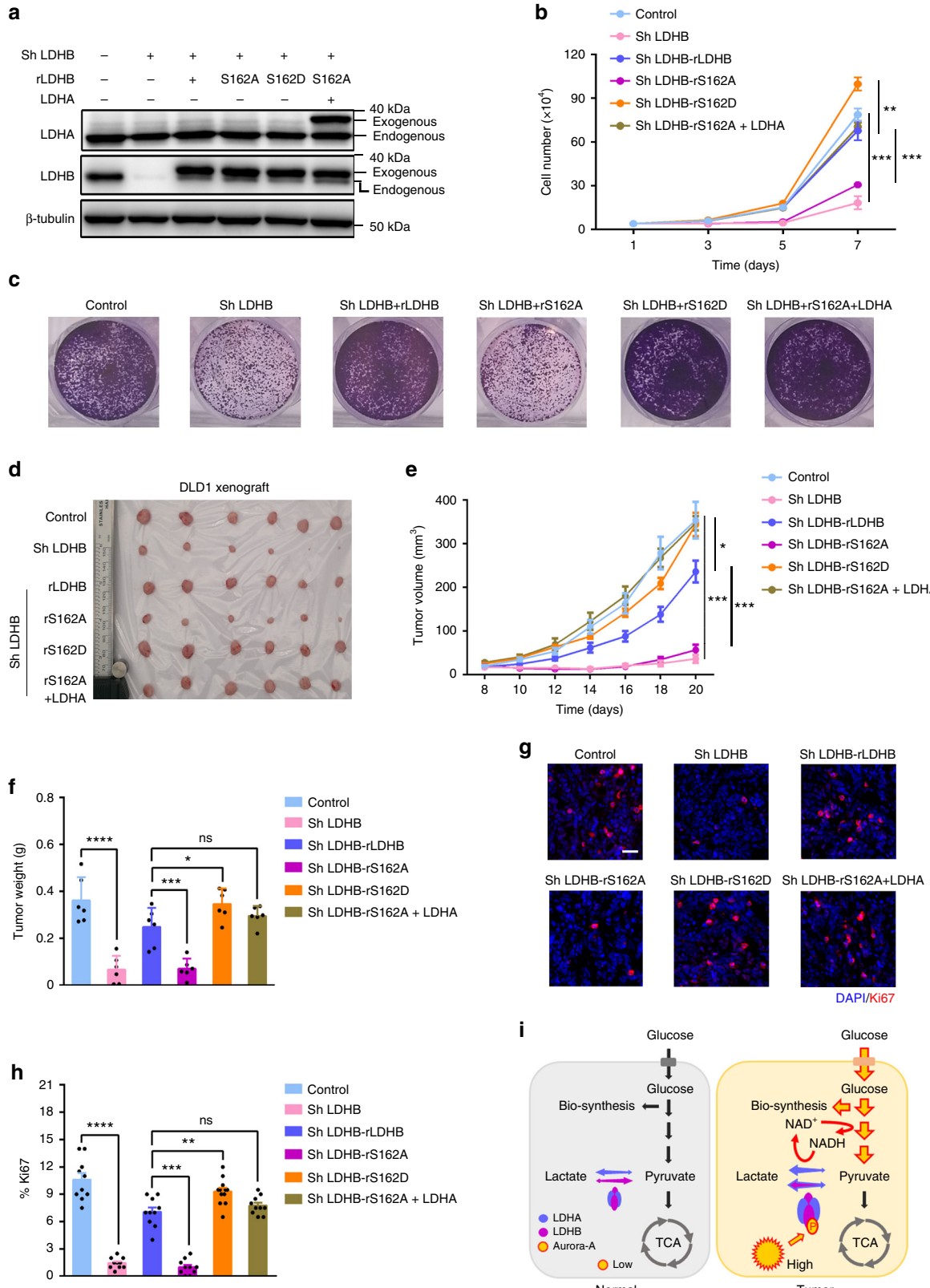

## Methods

**Cell culture and transfection**. Cells were cultured in a 5% $CO_2$ humidified incubator at 37 °C and were maintained in Dulbecco's modified Eagle's medium (Gibco, Grand Island, NY, USA) supplemented with 5% fetal bovine serum and 5% new born calf serum (Gibco, Grand Island, NY, USA), 100 units/ml penicillin and 100 μg/ml streptomycin (Beyotime Bio-technology, Jiangsu, China). Cell transfection was performed using Lipofectamine 3000 (Invitrogen, Carlsbad, CA, USA) or Polyethylen imine (Polysciences, Philadelphia, PA, USA).

**Cell lysis, IP, and western blotting**. Cells were lysed in a buffer containing 50 mM Tris-HCl (pH 7.4), 150 mM NaCl, 50 mM NaF, 1 mM EDTA, 1 mM $Na_2P_2O_4$, 1 mM $Na_3VO_4$, 1 mM PMSF and proteinase inhibitor cocktail (Sigma, St Louis, MO, USA). For Co-IP with FLAG-tagged proteins, cell lysate was incubated with anti-Flag M2-agarose (Sigma, St Louis, MO, USA) for 2 h at 4 °C; the beads were washed three times with TBS (50 mM Tris-HCl, 150 mM NaCl, pH 7.4), and the Flag-tagged proteins were eluted by Flag peptides (bank peptide, Hefei, China). For the Co-IP of endogenous proteins, whole-cell lysates were incubated with the

**Fig. 6** Phosphorylation of LDHB S162 promotes tumor progression. **a** In DLD1 cells endogenous LDHB was knocked down, then shRNA-resistant wild-type LDHB, LDHB S162A/D, and LDHA were expressed. The expression levels of LDHA/B were examined by WB. **b** Cell proliferation assay was conducted with the DLD1 cells tested in **a**. The growth curves were plotted over seven days. **c** Colony formation assay was conducted in DLD1 cells used in **a**. Cells were fixed after cultured for 10 days. Crystal violet staining of the cells were shown. **d** DLD1 cells tested in **a** were inoculated in nude mice. Xenograft tumors at the endpoint were collected and shown. **e** The growth curve of the tumors in **d**. **f** The weight of tumors at the endpoint in **d**. **g** Immuno-fluorescence staining of Ki67, a proliferation marker, in the sections of xenograft tumors in **d**, Scale bar, 50 μm. **h** Statistics of the index of the Ki67-positive cells in **g**. **i** A working model summarizes the major function of Aurora-A-LDHB pathway in the regulation of Warburg effect in cancer cells. The error bar in panels **b** represents the standard error of mean (SEM), $n = 3$ independent experiments. The error bar in panels **e**, **f**, **h** represents the standard deviation (SD), $n = 6$ biologically independent samples in panels **e**, **f** and $n = 10$ in panel **h**. Source data are provided as a Source Data files. (Student's $t$-test $*p < 0.05$, $**p < 0.01$, $***p < 0.001$, ns not significant).

corresponding antibody and protein A/G plus-agarose beads (Thermo Fisher Scientific, Inc., Waltham, MA, USA) overnight at 4 °C. The immunoprecipitates were washed three times with IP buffer (25 mM Tris, 150 mM NaCl, pH 7.2) and resuspended in loading buffer. For western blotting, immune-precipitates or whole-cell lysates were boiled at 95 °C for 5 min and separated on 10% SDS–polyacrylamide gels followed by transfer to polyvinylidene difluoride membranes (Millipore, Bedford, MA, USA). The membranes were blocked with 5% non-fat milk or 5% BSA in Tris-buffered saline containing 0.1% Tween 20 for 1 h and then incubated with primary antibodies and secondary antibodies. Detailed information for antibodies is provided in Supplementary Information. Signals were detected using Western Lightning Chemiluminescence Reagent Plus (Advansta, Menlo Park, CA, USA). Quantitative analysis was performed using Image J.

**Reagents and primers**. MLN8237 was purchased from Selleck (Shanghai, China; S1133) and was resolved in DMSO at 1 mM for stock. CoCl$_2$ was from (Sigma, St Louis, MO, USA; C8661), Oligomycin (Sigma, St Louis, MO, USA; 75351), 2DG (Sangon Biotech, shanghai, China; A602241), Puromycin (Sigma, St Louis, MO, USA; p8833), NADH (Sigma, St Louis, MO, USA; N8129), oxamate (Sangon Biotech, shanghai, China; A600871), Imidazole (Sangon Biotech, shanghai, China; A600799). Detailed information of primers used in this study is provided in Supplementary Information.

**Protein expression, purification, and ITC analysis**. BL21 *E. coli* were transformed with recombinant plasmids. Next, 50 μl of the transformed bacteria were plated onto LB agar containing 100 μg/ml ampicillin (amp), followed by incubation for 12 h at 37 °C. A single colony was inoculated into 50 ml of LB containing 100 μg/ml amp. The suspension was then shaken at 37 °C for 12 h, followed by the addition of 500 ml of fresh LB containing amp and continuation of growth until the OD reached 0.5–0.6. Next, 500 ml of fresh LB containing amp and 0.5 mM IPTG were added to induce protein expression at 16 °C for 24 h. The suspensions were then centrifuged at 6200 × g and 4 °C to collect the bacteria pellet. The pellets were resuspended in 30 ml of buffer (20 mM Tris, 50 mM KCl, pH 7.0). Next, the samples were subjected to ultrasonography to lyse the bacteria, followed by centrifugation at 18,000 × g and 4 °C to collect the supernatant containing His-tagged proteins. His-tagged protein was incubated with an NTA nickel column (Qiagen, Hilden, Germany) and eluted with 5 ml of 250 mM imidazole. Thereafter, the protein was further purified by gel filtration (Superdex 200 increase 10/300; GE Healthcare) in buffer containing 20 mM Tris, 50 mM KCl, pH 7.0. The peak fraction was collected and concentrated to 0.5 ml using an ultrafiltration tube (30 kDa). The protein concentration was quantitated with NanoDrop (Thermo Fisher Scientific, Waltham, MA, USA). Finally, the LDH solution was diluted to 40 μM in a buffer solution at pH 7.0 containing 20 mM Tris and 50 mM KCl. The NADH stock was 0.1 M and was diluted to 500–1000 μM before use. The oxamate stock was 20 mM and diluted to 1–2 mM before use. ITC experiments were conducted using a Microcal iTC200 microcalorimeter (GE Healthcare). The reaction cell contained 300 μl 40 μM LDH (or 300 μl 40 μM LDH preincubated with 200 μM NADH). Titrations were performed with every injection of 2 μl of titrant (NADH or oxamate) in the reaction cell, which was maintained at 25 °C. All the ITC data were analyzed by the Origin 7, and then followed by curve-fitting to one-site model to obtain the binding parameter[56].

**In vitro binding, in vitro kinase assay and autoradiography**. In vitro binding assay was performed as described previously[57]. For the in vitro kinase assay, the reaction was performed in 20 μl of 1 × kinase buffer (50 mM NaCl, 2 mM EGTA, 25 mM HEPES, pH 7.2, 5 mM MgSO$_4$, 1 mM DTT) containing 100 ng of human recombinant Aurora-A (Life Technologies Corporation, Carlsbad, CA, USA), 1 μg of his-tagged protein and 50 μM ATP. For the $^{32}$P assay, an additional 5 μCi of γ-$^{32}$P ATP was added into the kinase reaction mixture, followed by incubation for 30 min at 30 °C. The reactions were stopped with 5× loading buffer. The protein mixture was then heated at 95 °C for 5 min and separated by 10% SDS-PAGE. For the autoradiography assay, the gels were dried, followed by exposure to X-ray film.

**Quantification of the expression levels of LDHA/B in cell lines**. His-tagged recombinant LDHA and LDHB were expressed in *E. coli* and were purified by NTA nickel column (Qiagen, Hilden, Germany). The proteins were subjected to SDS-PAGE followed by Coomassie brilliant blue staining. The recombinant proteins were quantified based on the intensity of bands using the protein marker with known quantity (ProteinRuler II, TransGen Biotech, Beijing, China) as the reference. Next, the cell lysates were subjected to SDS-PAGE with quantified His-LDHA and LDHB and follow by WB using specific antibodies against LDHA and LDHB, respectively. Then the relative amount of endogenous LDHA and LDHB were quantified by comparing the intensity between His-tagged LDHA/B and endogenous LDHA/B based on the WB result using Image J. The relative expression levels of endogenous LDHA and LDHB in cell lines were then calculated according to the above results.

**LDHB knockdown and reintroduction**. The shRNA targeting human *LDHB* (5′-GGATATACCAACTGGGCTA-3′) was inserted into a GIPZ shRNA vector (Thermo Fisher Scientific, Waltham, MA, USA) according to the technical manual to obtain pGIPZ-shLDHB. To reintroduce *LDHB*, Flag-tagged human LDHB-WT, LDHB-S162A and LDHB-S162D containing five silent nucleotide substitutions in the sequence corresponding to the shRNA targeted region (5′-GG**T**TA**C**AC**A**AA**T**TGGGC**C**A-3′) were recombined into pGIPZ-shLDHB to replace GFP using a one-step cloning kit (Vazyme Biotech, Nanjing, China). Lentivirus was produced using a two-plasmid packaging system (pMD2.G and psPAX2). Briefly, 293 T cells were co-transfected with the pGIPZ vector expressing the shRNA sequence and resistant protein together with vectors expressing the gag and vsvg genes. The retroviral supernatant was harvested 48 h after initial plasmid transfection and was mixed with polybrene (8 μg/ml) to increase the infection efficiency. For transduction, sub-confluent DLD1, HeLa and U251 cells were infected with 3 ml of retrovirus for 24 h and then were selected in puromycin (DLD1 2 μg/ml, HeLa 0.5 μg/ml, U251 2 μg/ml) for 1 week.

**Measurement of LDHB enzyme activity and kinetics**. LDHB enzyme activity (lactate to pyruvate) was determined by using a Bioassay kit according to the manufacturer's instruction. LDHB enzyme activity (pyruvate to lactate) was conducted as described previously[8]. In brief, purified LDH was quantitated by Coomassie blue staining, then about 0.5 μg protein was added to reaction buffer containing 20 mM Tirs, 50 mM KCl, 1 mM NADH, and 10 mM pyruvate (pH 7.0). The change in absorbance (340 nm) resulting from NADH oxidation was measured using a Microplate reader (Thermo Fisher Scientific, Waltham, MA, USA)

The enzyme kinetic parameters were determined as previously described[58,59]. Briefly, the LDH activity was measured in the presence of pyruvate and NADH as substrates. Reactions were initiated by adding purified enzyme (0.5 μg) to a 100 μl total reaction volume containing 20 mM Tirs, 50 mM KCl, pyruvate (0–50 mM), and NADH (1 mM) in the 96-well microplates at 25 °C. Activity was monitored at 340 nm for checking the conversion between NADH and NAD$^+$ with a microplate reader.

**Metabolites analysis via LC/MS**. $1 \times 10^7$ DLD1 cells were washed twice with cold PBS, and polar metabolites were extracted by ice-cold 80% methanol immediately. Samples were sonicated after three rounds of freeze and thaw cycle with liquid nitrogen. After centrifugation, the supernatant was collected and dried. The pallet was dissolved in 80% methanol for LC-MS detection. For LC-MS analysis, an ExionLC™ UHPLC System combined with AB Sciex TripleTOF® 5600+ system was used and data acquisition via Analyst 1.7.1 software. A Phenomenex UHPLC KINETEX HILIC (150 cm × 2.1 mm, 2.6 μm) was used with mobile phase (A) consisting of 5 mM ammonium formate and 0.05% formic acid; mobile phase (B) consisting of 90% acetonitrile (ACN) and 10% water. Gradient program: mobile phase (A) was held at 10% for 5 min and then increased to 90% in 7 min; held for 2 min, then to 10% in 6 sec and held for 3 min before returning initial condition. The column was held at 40 °C and 5 μl of sample was injected into the LC-MS with a flow rate of 0.3 ml/min. Automatic calibrations of TOFMS were achieved with average mass accuracy of <2 ppm. Data Processing Software included Sciex PeakView 2.2, MasterView 1.1, and MultiQuant 3.0.2. Naturally occurring isotopes were corrected using Isocor software.

**Nano LC-MS/MS**. Flag-Aurora A was ectopically expressed in 293 T and DLD1 cells. Proteins were isolated by IP, and eluted using 200 µg/ml of flag peptide. The endogenous LDHB was enriched by IP and were subjected to SDS-PAGE. The gel was stained with Coomassie Brilliant Blue. Visualized bands were excised, de-stained with ammonium bicarbonate buffer, and dehydrated in 75% acetonitrile. Following rehydration (with 50 mM ammonium bicarbonate), the gel slices were crushed and subjected to overnight digestion with trypsin or chymotrypsin as described[60]. The peptides were extracted with acetonitrile containing 0.1% formic acid and vacuum dried. Proteolytic peptides were reconstituted with mobile phase A (2% acetonitrile containing 0.1% formic acid) and then separated on an on-line C18 column (75 µm inner diameter, 360 µm outer diameter, 10 cm, 3 µm C18). Mobile phase A consisted of 0.1% formic acid in 2% acetonitrile and mobile phase B was 0.1% formic acid in 84% acetonitrile. A linear gradient from 3 to 100% B over 75 min at a flow rate of 350 nL/min was applied. Mass spectrometry analysis was carried out on a Q-Exactive mass spectrometer (Thermo Fisher, SJ) operated in data dependent scan mode. Survey scan ($m/z$ 375–1300) was performed at a resolution of 60,000 followed by MS2 scans to fragment the 50 most abundant precursors with collision induced dissociation. The activation time was set at 30 ms, the isolation width was 1.5 amu, the normalized activation energy was 35%, and the activation q was 0.25. Mass spectrometry raw file was searched by Proteome Discovery version 1.3 using MASCOT search engine with percolator against the human ref-sequence protein database (updated on 07–04–2016). The mass tolerance was set to be 20 ppm for precursor and 0.5 Da for product ion. Missed cleavages were no more than two for each peptide. Phosphorylation of Ser/Thr and Tyr, Oxidation (Met), Acetyl (N terminus) were used as variable modifications. A filter of 90% peptide confidence was applied according to the PeptideProphet and ProteinProphet parsimony algorithms. Fragment assignment of each modified peptide was subject to manual inspection and validation using the original tandem mass spectra acquired in profile mode. The phosphorylation position was validated using the ptmRS algorithm. Phosphopeptide abundances were further normalized against the abundance of total peptides for LDHB[61–63]. The relative abundance was then compared between different conditions.

**Cell synchronization and flow cytometry**. Hela cells were synchronized with a double thymidine block procedure. Briefly, the exponentially growing Hela cells were maintained with 2 mM thymidine for 18 h, followed by a release of 9 h in fresh medium, and then cells were re-cultured in 2 mM thymidine for additional 15 h. Highly synchronized cells at G1 phase were obtained. Then after a release of 4 h in fresh medium, the cell population entered into S phase. After a release of 9 h in fresh medium, the cells entered into M phase. Then mitotic cells were harvested by mitotic shake-off. The cells remained attached after shaking-off were collected as cells at G2 phase. For cell cycle analysis, cells were trypsinized or shaked off. Then, Cells were washed with cold phosphate-buffered saline (PBS), fixed by cold 70% ethanol, and suspended in a staining solution, which contains 20 µg/ml of propidium iodide, 0.1% Triton X-100 and 200 µg/ml of RNaseA, followed by analysis using a FACScan (BD Biosciences) and Modfit 2.0.

**Structure analysis and molecular dynamics simulation**. The complex structure of LDHB and the coenzyme NADH (PDB entry 1i0z) was used to setup dynamics (MD) simulations. The first one is for the wild-type structure (WT). In the second simulation and S162 was phosphorylated (pS162). MD simulations were performed with a parallel implementation of the GROMACS-4.5.5 package, using the CHARMM27 force field. Taking WT as an example, the setup procedure was as follows. The periodic boundary condition (PBC) with a dodecahedron box type was used, with the minimum distance between the solute and the box boundary of 1.2 nm. The box was filled with TIP3P water molecules[64]. The system with the solute (including the protein and NADH) and waters was energy-minimized by the steepest descent method, until the maximum force was smaller than $1000 \, kJ \, mol^{-1} \, nm^{-1}$. Replacing the water molecules at the positions with the most favorable electrostatic potential added in 34 K$^+$ and 26 Cl$^-$ to compensate for the net negative charge of the solute and to mimic the salt concentration (50 mM). The final system was energy-minimized again using the steepest descent and then the conjugate gradient method, until the maximum force was smaller than $400 \, kJ \, mol^{-1} \, nm^{-1}$. The simulation was conducted by using the leap-frog algorithm[65] with a 2 fs time-step. Before the production run, a 50-ps equilibration simulation with positional restraint was carried out, using a force constant of $1000 \, kJ \, mol^{-1} \, nm^{-2}$. The initial atomic velocities were generated according to a Maxwell distribution at 300 K. The simulation was performed under the constant NPT condition. The three groups (solute, solvent, and ions) were coupled separately to a temperature bath of 300 K by using an velocity rescaling thermostat[66], with a relaxation time of 0.1 ps. The pressure was kept at 1 bar with a relaxation time of 0.5 ps and the compressibility of $4.5 \times 10^{-5} \, bar^{-1}$. Covalent bonds in the protein were constrained using the P-LINCS algorithm[67]. Twin-range cutoff distances for the van der Waals interactions were chosen to be 0.9 and 1.4 nm, respectively, and the neighbor list was updated every 20 fs. The long-range electrostatic interactions were treated by the PME algorithm[68], with a tolerance of $10^{-5}$ and an interpolation order of 4. A 50 ns production run was conducted. For pS162, the setup procedure was nearly the same as WT. Hydrogen bond dynamics was calculated with modeling data and plotted over time.

**LDH isoenzyme characterization**. Native gel electrophoresis was used to separate and characterize the five known LDH isoenzymes. The assay was conducted as described previously[69]. Samples (20 µg protein) were loaded into a 1.2% agarose gel in 25 mM Tris·HCl and 250 mM glycine (pH 9.5) buffer, with a 6× loading buffer consisting of 0.1% bromophenol blue, 15% glycerol, and 20 mM Tris·HCl (pH 8.0). Electrophoresis was conducted for 240 min at 110 V in a 5 mM Tris·HCl and 40 mM glycine (pH 9.5) running buffer. Gels were then washed briefly in 100 mM Tris·HCl (pH 8.5) buffer. To visualize LDH isoenzyme bands, the gel was incubated for 20 min at 37 °C in a solution containing lactate (3.24 mg/mL), β- nicotinamide adenine dinucleotide (0.3 mg/mL), NBT (0.8 mg/mL), and PMS (0.167 mg/mL) dissolved in 10 mM Tris·HCl (pH 8.5) buffer.

**Metabolites analysis**. The metabolites (Glucose, lactate, G6P, 2PG, NADPH, and GSH) were measured with kits (BioVision technologies, Milpitas, CA, USA; Bioassay systems, Hayward, CA, USA) following the manufacturer's instruction. The values were normalized to the cell number. To determine the absolute intracellular concentration of pyruvate, cells were washed twice with PBS and harvested after trypsinization. The suspension was centrifuged at $100 \times g$ for 5 min and the supernatant was removed. Cells was resuspended with PBS, centrifuged at $100 \times g$ for 5 min. Repeat twice, then marker the total cell volume(V1). Lysate cells with lysis buffer at 4 °C for 10 min. Centrifuged at $18,000 \times g$ and 4 °C for 10 min. Marker the total lysis volume(V2). Determinate the pyruvate concentration in the cell lysis supernatant (C1) immediately with BioVision kit (BioVision technologies, Milpitas, CA, USA). Calculate intracellular pyruvate concentration(C) with following formula: C = C1 × V2/V1.

**Living cell imaging for NADH/NAD$^+$ biosensor**. The Sonar biosensor was transfected into DLD1 cells and cells were plated on a 35-mm glass-bottom dish with pyruvate free DMEM medium. Images were acquired with a Delta Vision microscope (GE Healthcare, Buckinghamshire, UK) equipped a ×60 objective lens, NA = 1.42. Detailed procedures was performed as described previously[27].

**Fluorescence resonance energy transfer (FRET)**. DLD1 cells were transiently transfected with Clover-LDH and mRuby2-Aurora-A using Lipofectamine 3000 according to the manufacturer's protocol. Cells were treated with 200 µM CoCl$_2$ for 10 h before fixed with 4% PFA at 36 h after transfection. Images were acquired with a Delta Vision microscope (GE Healthcare, Buckinghamshire, UK) equipped a ×60 objective lens, NA = 1.42. For FRET acquisition, three channels were defined as follows: channel 1 for donor only (donor excitation with donor emission); channel 2 for FRET (donor excitation with acceptor emission); and channel 3 for acceptor only (acceptor excitation with acceptor emission). Three channels were taken with equal exposure times (100 ms). The FRET efficiency was calculated as described previously[70].

**ECAR and OCR measurement**. $2 \times 10^4$ cells were plated per well in XF 96-well microplates and incubated for 24 h at 37 °C in 5% CO$_2$ or seeded in pre-coated XF 96-well microplates and centrifuged at $100 \times g$ for 5 min just before assay. The ECAR and OCR were measured with an XF96 Extracellular Flux Analyzer (Seahorse biosciences, North Billerica, MA, USA) as the manufacturer's protocol.

**Cell proliferation and clone formation**. $2 \times 10^4$ cells were seeded in triplicate in plates and cell numbers were counted every day over a 6-day period for DLD1 cells. For clone formation, $1 \times 10^4$ cells were seeded in six-well culture dishes and cells were fixed in 4% PFA and stained with 0.2% crystal violet ten days later.

**Xenograft**. Briefly, $3 \times 10^6$ DLD1 stable cell lines (Hela and U251, $5 \times 10^6$) were collected and resuspended in 50 µl PBS followed mixed with 50 µl Matrigel, and then injected subcutaneously into the left or right flank of 5-week-old male BALB/c nude mice purchased from SHANGHAI SLAC LABORATORY ANIMAL COMPANY. Tumor formation was assessed every 2–3 days. Tumor growth was measured with calipers at the indicated time, starting on day 7 after the injection. Tumor growth was recorded by measuring two perpendicular diameters of the tumors over a 4-week time course according to the formula Tumor volume (mm$^3$) = length (mm) × width (mm) × height (mm) × 0.52. After 4 weeks, the mice were killed, followed by isolation of xenograft whose weights and volumes were determined at the experimental endpoint. Statistical analysis of tumor volumes and weights were performed using a paired Student's $t$-test. All animals were kept in specific pathogen-free conditions. All experimental procedures involving mice were carried out as prescribed by the National Guidelines for Animal Usage in Research (China) and were approved by the Ethics Committee at the University of Science and Technology of China.

**IHC staining**. Colon tumor samples were acquired from the Department of Oncology, the First Affiliated Hospital of Anhui Medical University. Tissues were fixed in 10% buffered Formalin for 6 h, followed by transfer to 70% alcohol. These paraffin-embedded tissues were sectioned (4 mm) and stained with hematoxylin. Antigen retrieval was performed by incubating the slides in 10 mM citric acid buffer (pH 6.0) or Tris-EDTA buffer (10 mM Tris Base, 1 mM EDTA, pH 9.0) at

95 °C for 15 min. The endogenous peroxidase activity was inactivated in a solution containing 3% hydrogen peroxide ($H_2O_2$) in methanol. The following detection and visualization procedures were performed according to the manufacturer's protocol (ZSGB-BIO, Beijing, China).

**Ethics statement.** Samples were obtained with informed consent and all protocols were approved by The First Affiliated Hospital of Anhui Medical University. Written informed consent was obtained from all patients.

**Statistical analysis.** Data were from three or more independent experiments. Statistical analyses were carried out by GraphPad Prism 5 software (La Jolla, CA, USA). The data were presented as mean ± SD or mean ± SEM. Student's $t$-test was used to calculate p-values. Statistical significance is displayed as $*P < 0.05$, $**p < 0.01$. $***p < 0.001$, $****p < 0.0001$, ns: not significant.

**Reporting summary.** Further information on research design is available in the Nature Research Reporting Summary linked to this article.

## Data availability

Datasets containing expression and clinical data were accessed from the GEO website under the accession codes GSE39582 (colon), GDS3257 (lung) and GSE63514 (cervical cancer). The mass spectrometry proteomics data have been deposited to the ProteomeXchange Consortium via the PRIDE partner repository with the dataset identifier PXD016110 (Fig. 2a). The source data underlying Figs. 1a-1h, 2a-h, 2k-l, 3b-f, 3h, 3j, 4c-d, 5a-h, 6a-b, 6e-f, 6h and Supplementary Figs. 1a-j, 2a-c, 2g-h, 2j, 2l-n, 3a-j, 4d, 5a-e, 6b-d, 6f-h, 6j-n, 6p-q are provided as source Data file.

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

## Acknowledgements

We thank Dr. Yi Yang and Dr. Yuzheng Zhao (Synthetic Biology and Biotechnology Laboratory, East China University of Science and Technology) for providing the SoNar biosensor. We thank all members of the Yang group for the discussions. This work was supported by the Strategic Priority Research Program of the Chinese Academy of Sciences, Grant No. XDB19000000, grants from the Ministry of Science and Technology of China (2015CB553804 to ZY), National Science Foundation of China (31701178, 31970670 to J.G.; 31571396, 31771498 to Z.Y.; 21573205 to Z.Z. and 81673387 to X.D.), the Fundamental Research Funds for the Central Universities to Z.Y. and J.G., the Beijing Nova Program Z161100004916163 to XD and a National Thousand Young Talents Award to Z.Y. This work was also supported in part by the Open Project of the CAS Key Laboratory of Innate Immunity and Chronic Disease to Z.Y., and the New Concept Medical Research Fund to Y.P.

## Author contributions

A.C., P.Z., J.G. and Z.Y. conceived and designed the study. Z.Y. and J.G. supervised the project. D.Y. conducted the glycolytic flux assay and mass spectrum analysis of metabolites. Y.J. performed IP experiments. B.W., X.D., X.P., Z.S. and G.W. conducted mass spectrum identification of protein and modification. A.C., T.X. and J.S. performed enzymatic activity measurement. Y.J. and C.Z. performed ITC and crystallographic calculation. A.C., P.Z. and J.S. conducted plasmids construction and western analysis. Z.Z. and C.D. did the structural simulation. M.W. performed autophagy analysis. H.Z. and P.G. did the hypoxia experiment. T.X., Q.W., H.W. and Y.P. did the clinical data analysis. A.C. did the FRET and biosensor assay. A.C., P.Z., J.G. and Z.Y. performed statistical analysis. A.C., J.G. and Z.Y. performed overall data interpretation and wrote the paper.

## Competing interests

The authors declare no competing interests.
