## [Peer Review File · Nature Communications]

Reviewers' Comments:

Reviewer #1:

Remarks to the Author:

In their manuscript, A Cheng et al. report that serine/threonine protein kinase Aurora-A interacts with lactate dehydrogenase B (LDHB) and phosphorylates LDHB on S162 in cancer cells. Using a well-adapted strategy and methods at the state of the art, the authors convincingly demonstrate that LDHB S162-phosphorylation relieves substrate inhibition of the enzyme by pyruvate, thereby reversing the normal function of the enzyme towards the reduction of pyruvate to lactate. Thus, through its action on LDHB, Aurora-A promotes aerobic glycolysis, cancer cell proliferation and tumor growth in mice. The study is original and its most of its conclusions are strongly grounded on experimental observations. Still, some comments should be addressed:

Major comments:

1 – I do not understand how an increase in the reduction of pyruvate to lactate upon LDHB S162-phosphorylation increases the shunting of carbon intermediates from glycolysis to side pathways, such as the PPP. LDHB S162-phosphorylation should rather promote glycolytic ATP production, i.e., in a same process as PKM2 activation. When PKM2 is inhibited, it promotes biosynthesis from glycolysis, and when PKM2 is inactivated, it promotes ATP production from glycolysis (Mazurek S et al. *Int J Biochem Cell Biol* 2011;43:969-80). Similarly, when LDHB is phosphorylated it should promote ATP production and when dephosphorylated it should promote biosynthesis. Please explain, by also showing the glucose/lactate ratio before and after LDHB phosphorylation in your cell models.

2 - What are the effects of LDHB phosphorylation on cell respiration (OCR)?

3 – Is the control of LDHB by Aurora-A conserved under hypoxia? This experiment is key to determine whether aerobic and anaerobic glycolysis similarly exploit LDHB phosphorylation. The experiment with CoCl₂ activation HIF-1 does not fully reproduce cancer cell adaptation to hypoxia.

4 – Please explain how Aurora-A binding with its catalytic site to amino-acids 199-250 of LDHB can promote the phosphorylation of S162, which is rather distant in the LDHB sequence. If the catalytic site of Aurora-A is bound to aa 199-250, how can it phosphorylate S162? Can the flexibility of the region bearing S162 be modeled?

5 – Based on the experiment on Fig. 3h, it would be great to adapt the experimental conditions to further show an oxidation of NADH to NAD⁺ by phosphorylated LDHB.

6 – Authors insist on the p53 status of their cell models. Please discuss the role of p53 in the Aurora-A-LDHB system.

Minor comments:

1 – In several sentences, there is a confusion between 'in vivo' and 'in vitro'. If authors want to insist on the fact that some observations are made on intact cells in vitro, they can use 'in cellulo' instead of 'in vivo'.

2 – Page 6. The list of Aurora-A binding proteins should be shown in supplemental data.

3 – Statistics are missing in Figure 2k. Please also show the quantification of the phosphorylation of the other residues in supplemental data.

4 – 'Conversion of' could be favorably replaced by 'oxidation of' or 'reduction of' where convenient

throughout the manuscript.

Reviewer #2:

Remarks to the Author:

The manuscript "Aurora-A phosphorylation of lactate Dehydrogenase B.....substrate inhibition effect" describes interesting findings on the involvement of Aurora-A in regulating glycolytic flux by phosphorylating LDHB subunit of the tetrameric LDH enzyme, which may have significant implications on the role of Aurora-A in inducing Warburg effect in cancer cells. Although the results presented are compelling but a more robust experimental strategy with more cancer cell lines is required to convince the readers about the validity of the conclusions drawn. Also, the data needs to be presented in a more organized manner and the text of the manuscript requires major editing along with correction in the citation of references at multiple places. Some of these concerns are mentioned below:

1. Details of the Mass Spectrometry data indicating Aurora-A interaction with LDH B in terms of the number of hits with the corresponding peptides should be mentioned. Are the identified peptides from the binding domain mapped in Fig 2 e, f and not conserved in LDHA? This is important since the co-IP experiment with Flag tagged constructs shows Aurora-A binding with LDHA also, albeit less than that with LDHB.
2. The co-IP experiment of endogenous protein should include reciprocal experiment of immunoprecipitation with LDHB/LDHA abs and should be shown with more cancer cell lines with varying levels of expression of Aurora-A and LDH. It is intriguing that the endogenous binding data are shown for cells with inactivated p53 (DLD1 and 293T) but authors do not discuss if functional status of p53 may have any relevance to their findings while also mentioning that expression of Aurora-A and LDHB show significant positive correlation in colon cancer samples with p53 mutation (page 10).
3. The recombinant GST-Aurora-A enzyme produced in E.coli does not have the conformation like that in mammalian cells, so the in vitro relative enzyme activity assays for forward and reverse reactions in presence of recombinant GST-Aurora-A (Fig 3f) needs to be done with appropriate recombinant enzymes.
4. The ITC and the modeling assays indicating that the activity of the forward reaction is compromised when LDHB S162 is phosphorylated and that the effect is rescued due to the phosphorylated form relieving pyruvate substrate inhibition raises additional possibilities, which should be carefully investigated. For example, since the LDHB S162 phosphorylated subunit still remains in the tetrameric form, does LDHA may still be important in driving the forward reaction by the enzyme with phosphorylated LDHB?
5. The text of the manuscript needs major editing. For example, the sentence "Thus, the current data support a model.....glycolytic flux and tumor growth" and the following sentences on page 20 do not make much sense in the context, as stated. References also need correction at several places.

Reviewer #3:

Remarks to the Author:

Using cell culture experiments and Xenograft models Cheng et al. examine the involvement of Aurora A in the regulation of glycolysis in cancer cells by phosphorylating Lactate Dehydrogenase B (LDHB). The authors claim that Aurora A phosphorylates LDHB at Serine 162 driving the balance towards the forward reaction that converts pyruvate to lactate. The changes in LDHB enzymatic activity regulate the glycolytic influx and NAD⁺ levels causing a rapid tumor progression in multiple cancers. Furthermore, the expression of Aurora A and LDHB are significantly upregulated in various tumors indicating that both proteins are associated with cancer progression in human patients. Overall the results achieved in this study may be of interest to the field of and could broaden our understanding of metabolism regulation in cancer cells. The manuscript represents an

impressive amount of work. However, before publication several key issues need to be addressed:

Major points:

1- Figure 2j: The authors claim that S162 is the main phosphorylation site of Aurora A in LDHB based on MS data. It is not clear which peaks represent S162 +/-phosphorylation. Additional experiments need to be included to verify this statement. One of the important papers that has analyzed the consensus site for Aurora A was published by Kettenbach et al. in Science Signaling (28 June 2011). This paper reports the stringent consensus site to be RRXp(S/T) or the less stringent sites to be -2 (RXpS/T) or -3 (RXXp(S/T)). S126 as shown in Fig. 2m does not match with all three consensus sites (stringent and non-stringent) that were the result from 778 phosphorylation sites on 562 proteins analyzed by Kettenbach et al.

Therefore, the use of an unspecific pan-phospho Serine antibody to confirm the phosphorylation on S162 by Aurora A is not sufficient. Since the entire paper is based on the claim that Aurora A phosphorylates LDHB at S162 and this phosphorylation regulates glycolysis and tumor growth, there is an urgent need for a phospho-specific antibody that recognizes pS162 as discussed by the authors on page 25.

2- Since the enzymatic activity of Aurora A is cell-cycle regulated with a peak in G2/M, a kinetics of LDHB phosphorylation (S162) in synchronized cells is essential to answer the question whether the phosphorylation of S162 is cell cycle regulated.

3- Another question that remains unanswered is whether Aurora A is the only protein kinase that might phosphorylate S162. This aspect is very important because the inhibition of Aurora A using MLN8237 shows only a relatively small downregulation of the glycolytic rates (Fig. 1d). The authors used the Aurora A inhibitor MLN8237 at a concentration of 200 nM. At this relatively high concentration the ATP-competitive inhibitor MLN8237 might also inhibit other protein kinases. Therefore, the phosphorylation of S162 should also be tested under Aurora A depletion by RNAi.

4- A major weakness of the manuscript is that the phosphorylation of S162 was never tested in endogenous LDHB. Expression and phosphorylation of recombinant protein might not reflect the real situation.

5- Figure 2. The authors reported the presence of many Aurora A G2/M binding partners (Bora, TPX2) along with LDHB in the IP precipitates. Unless LDHB binds to ARKA during G2/M, this observation is likely due to the fact that these IPs were performed in unsynchronized cells, where the G2/M fraction of cells is somehow enriched. It will be very important to investigate whether the association of Aurora and LDHB is cell cycle -dependent and in which cell cycle phase this interaction takes place. A thymidine synchronization followed by a release of cells should be performed here.

6- Is LDHB regulated in a cell-specific manner?

7- Suppl. Figure 2. The FRET assays have been performed in randomly taken cells without any synchronization. An analysis in mitotic and non-mitotic cells is required. Single controls are missing (mRuby-Aurora A and single Clover-LDHA)

Minor points

8- Figure 2j: The meaning of mass spectrometry data/peaks should be explained in the result part and the legend is too short.

9- Figure 1: For the sake of the general reader more explanation of the ECAR assays should be provided. The figure legends are kept too short.

10- Suppl Figure 4c and d: require more explanation within results as well as in legends

11- In the introduction the authors should include a paragraph, where they describe the activation of Aurora A in and outside of mitosis.

We are very grateful for those constructive comments from the reviewers and have taken these criticisms with great attention. Now we have addressed all these issues with further work and more accurate statement, which significantly improve the overall quality of the manuscript and make conclusions much more solid.

Listed below are the point-by-point response (*in blue italicized font.*) to the reviewers' critiques.

All the changes in the text of manuscript are highlighted as red font.

Reviewers' comments:

Reviewer #1: Cancer metabolism
(Remarks to the Author):

In their manuscript, A Cheng et al. report that serine/threonine protein kinase Aurora-A interacts with lactate dehydrogenase B (LDHB) and phosphorylates LDHB on S162 in cancer cells. Using a well-adapted strategy and methods at the state of the art, the authors convincingly demonstrate that LDHB S162-phosphorylation relieves substrate inhibition of the enzyme by pyruvate, thereby reversing the normal function of the enzyme towards the reduction of pyruvate to lactate. Thus, through its action on LDHB, Aurora-A promotes aerobic glycolysis, cancer cell proliferation and tumor growth in mice. The study is original and its most of its conclusions are strongly grounded on experimental observations. Still, some comments should be addressed:

We thank the reviewer for his/her enthusiasm and insightful summary for this study.

Major comments:

1 – I do not understand how an increase in the reduction of pyruvate to lactate upon LDHB S162-phosphorylation increases the shunting of carbon intermediates from glycolysis to side pathways, such as the PPP. LDHB S162-phosphorylation should rather promote glycolytic ATP production, i.e., in a same process as PKM2 activation. When PKM2 is inhibited, it promotes biosynthesis from glycolysis, and when PKM2 is inactivated, it promotes ATP production from glycolysis (Mazurek S et al. Int J Biochem Cell Biol 2011;43:969-80). Similarly, when LDHB is phosphorylated it should promote ATP production and when dephosphorylated it should promote biosynthesis. Please explain, by also showing the glucose/lactate ratio before and after LDHB phosphorylation in your cell models.

This is a very reasonable concern. The word “shunting” was misused in the original manuscript, which caused the misleading.

In this study, we demonstrated Aurora-A mediated LDHB S162 phosphorylation relieves the substrate inhibition effect, resulting in promotion of the reduction of pyruvate to lactate. Meanwhile, NAD⁺ regeneration is promoted (Fig. 3 and Supplementary Fig. 5b), which significantly enhances the overall glucose influx (Fig. 5g and 5h) and bio-synthesis with glycolytic intermediates (Fig. 5g, 5h and Supplementary 5a, 5d, 5e). Therefore, in the original manuscript, we proposed that when glycolytic flux is significantly increased, lactate production and anabolic biosynthesis are promoted at the same time after Aurora-A mediated LDHB phosphorylation (see the second paragraph on page 18, the first paragraph on page 20 and the first paragraph on page 21). Also see the model below and description in the last paragraph of the results.

The role of LDHB phosphorylation is similar as that of LDHA upregulation in glucose metabolism (reference 9, 53 of the manuscript), but is different from the function and regulation of PKM2.

So we did not intend to compare the ratio of glycolytic intermediates that go through the biosynthesis before and after LDHB phosphorylation. Neither would we propose any rewiring from lactate production to biosynthesis in utilizing glycolytic intermediates. But we used the wrong word “shunting”, which caused the misunderstanding. Now, the statements of these data has been rephrased accordingly.

Thanks so much for raising this issue.

2 - What are the effects of LDHB phosphorylation on cell respiration (OCR)?

This is a very good suggestion.

The OCR has been analyzed in LDHB WT and S162D cells with/without Aurora-A inhibitor (Supplementary fig. 5c). The result shows that there is little difference in oxygen consumption before and after LDHB phosphorylation. Therefore, LDHB S162 phosphorylation does not affect oxygen consumption.

3 – Is the control of LDHB by Aurora-A conserved under hypoxia? This experiment is key

to determine whether aerobic and anaerobic glycolysis similarly exploit LDHB phosphorylation. The experiment with CoCl₂ activation HIF-1 does not fully reproduce cancer cell adaptation to hypoxia.

We agree with the review's comment.

The association of LDHB with Aurora-A and the phosphorylation of LDHB S162 have been examined in cells that are cultured in hypoxic incubator (1% oxygen). Consistent with the data obtained after CoCl₂ treatment (Supplementary fig. 2h), the interaction between Aurora-A and LDHB was somewhat enhanced in hypoxia (Supplementary fig. 2g). The phosphorylation of LDHB S162 was also slightly increased in hypoxia condition (Supplementary fig. 2o). Therefore, Aurora-A mediated regulation of LDHB occurs under both aerobic and anaerobic environment.

4 – Please explain how Aurora-A binding with its catalytic site to amino-acids 199-250 of LDHB can promote the phosphorylation of S162, which is rather distant in the LDHB sequence. If the catalytic site of Aurora-A is bound to aa 199-250, how can it phosphorylate S162? Can the flexibility of the region bearing S162 it be modeled?

The catalytic domain of Aurora-A is relatively big (~250 amino acids long, 123-388aa). It is possible that there are two sites that mediate the interaction of Aurora-A and LDHB. The domain 199-250 of LDHB mediates a interaction with Aurora-A c-terminal catalytic domain, but does not bind with the catalytic center of Aurora-A. Another domain including serine 162 mediates a weak interaction with Aurora-A catalytic center, which facilitates the phosphorylation of S162.

It is difficult to model the interaction interface of LDHB-Aurora-A complex without precise binding sites. There are so many possibilities in the model with the current information.

5 – Based on the experiment on Fig. 3h, it would be great to adapt the experimental conditions to further show an oxidation of NADH to NAD⁺ by phosphorylated LDHB.

This is very good suggestion. The experiment has been done and the results indicated that the activity of forward reaction is increased after overexpression Aurora-A, which promotes the phosphorylation of LDHB (Fig. 3i, j and supplementary Fig. 5b).

6 – Authors insist on the p53 status of their cell models. Please discuss the role of p53 in the Aurora-A-LDHB system.

This is also very good suggestion. We now provided insights of p53 and Aurora-A-LDHB signals in the first paragraph of the discussion (page 21).

Minor comments:

1 – In several sentences, there is a confusion between 'in vivo' and 'in vitro'. If authors want to insist on the fact that some observations are made on intact cells in vitro, they can use 'in cellulo' instead of 'in vivo'.

Thanks very much for the comments. We have made the changes accordingly.

2 – Page 6. The list of Aurora-A binding proteins should be shown in supplemental data.
The information has been added to supplemental table 1.

3 – Statistics are missing in Figure 2k. Please also show the quantification of the phosphorylation of the other residues in supplemental data.
The statistical data has been included for the new data.

4 – ‘Conversion of’ could be favorably replaced by ‘oxidation of’ or reduction of’ where convenient throughout the manuscript.
Thanks for the suggestion. We have changed the phrases accordingly.

Reviewer #2: Cancer signalling

(Remarks to the Author):

The manuscript “Aurora-A phosphorylation of lactate Dehydrogenase B.....substrate inhibition effect” describes interesting findings on the involvement of Aurora-A in regulating glycolytic flux by phosphorylating LDHB subunit of the tetrameric LDH enzyme, which may have significant implications on the role of Aurora-A in inducing Warburg effect in cancer cells. Although the results presented are compelling but a more robust experimental strategy with more cancer cell lines is required to convince the readers about the validity of the conclusions drawn. Also, the data needs to be presented in a more organized manner and the text of the manuscript requires major editing along with correction in the citation of references at multiple places.

We thank the Reviewer for his/her insightful comments and criticisms.

Some of these concerns are mentioned below:

1. Details of the Mass Spectrometry data indicating Aurora-A interaction with LDH B in terms of the number of hits with the corresponding peptides should be mentioned. Are the identified peptides from the binding domain mapped in Fig 2 e, f and not conserved in LDHA? This is important since the co-IP experiment with Flag tagged constructs shows Aurora-A binding with LDHA also, albeit less than that with LDHB.

This is a reasonable concern.

The detailed information of Mass Spectrometry analysis has been included in the results and supplementary tables. Five peptides of LDHB were identified in the MS analyses. Four out of five peptides exclusively belong to LDHB. One peptide is conserved in LDHA (Supplementary Table 2). However, all the LDHB peptides identified in the MS analyses

are not within the binding domain, because it could be any peptides within LDHB after proteolysis.

Co-IP data demonstrated the interaction between Aurora-A and LDHA is much weaker than the interaction between Aurora-A and LDHB (Fig. 2c, d). In FRET assay, no direct interaction was detected between Aurora-A and LDHA (Supplementary Fig. 2e). The domain that mediates the interaction between LDHB and Aurora-A is also different from the corresponding region in LDHA (Supplementary fig. 2i). These data support that Aurora-A modulates LDH through interacting with LDHB. The LDHA in the precipitant of Co-IP assay is probably from the LDH tetramer containing both LDHB and LDHA subunits.

2. The co-IP experiment of endogenous protein should include reciprocal experiment of immunoprecipitation with LDHB/LDHA abs and should be shown with more cancer cell lines with varying levels of expression of Aurora-A and LDH. It is intriguing that the endogenous binding data are shown for cells with inactivated p53 (DLD1 and 293T) but authors do not discuss if functional status of p53 may have any relevance to their findings while also mentioning that expression of Aurora-A and LDHB show significant positive correlation in colon cancer samples with p53 mutation (page 10).

Reciprocal co-IP has been done using LDHB antibody (Supplementary fig.2c). Consistently, Aurora-A was detected in the LDHB precipitant in DLD1, 293T and U251 cells, but not in RKO cells (Supplementary fig.2c)

As suggested, we discuss the correlation of p53 status and Aurora-A-LDHB signaling in the discussion (first paragraph in page 21).

3. The recombinant GST-Aurora-A enzyme produced in E.coli does not have the conformation like that in mammalian cells, so the in vitro relative enzyme activity assays for forward and reverse reactions in presence of recombinant GST-Aurora-A (Fig 3f) needs to be done with appropriate recombinant enzymes.

In the previous experiment, the recombinant GST-Aurora-A was allosteric activated by TPX2 peptide (1-25 aa), which is now proved by the phosphorylation of Histone H3 (a known substrate, supplementary Fig. 3b). Therefore, with the support from TPX2 peptide, the recombinant GST-Aurora-A has the active conformation (Molecular mechanism of Aurora A kinase auto-phosphorylation and its allosteric activation by TPX2. eLife.2014).

4. The ITC and the modeling assays indicating that the activity of the forward reaction is compromised when LDHB S162 is phosphorylated and that the effect is rescued due to the phosphorylated form relieving pyruvate substrate inhibition raises additional possibilities, which should be carefully investigated. For example, since the LDHB S162 phosphorylated subunit still remains in the tetrameric form, does LDHA may still be important in driving the forward reaction by the enzyme with phosphorylated LDHB?

Thanks so much for this very important question.

We have tested this hypothesis by new experiment. When LDHA was knocked down in the

LDHB S162D expressing cells, the activity of forward reaction of LDH was not significantly compromised (Supplementary Fig.3f), suggesting that LDHB S162D alone can efficiently catalyze the reduction of pyruvate to lactate.

In addition, in LDHB S162A expressing cells, the forward reaction is significantly compromised in the presence of LDHA in the tetramer (Fig. 3c). Overexpressing Aurora-A increased the activity of WT LDHB but not LDHB S162A in the presence of LDHA (Fig. 3e). These results further supported that LDHA alone is not sufficient to maintain a high activity of the forward reaction, though it is in the LDH tetramer. Aurora-A mediated phosphorylation of LDHB S162 significantly contributes to the increase in reduction of pyruvate to lactate.

Furthermore, the enzymatic analysis unequivocally demonstrated the activity of the forward reaction of LDHB is significantly promoted in recombinant proteins, IP purified proteins from cell lysate and proteins in the live cells (Fig.3).

Therefore, we propose that the increase of the catalytic activity of LDHB in reduction of pyruvate to lactate upon phosphorylation at serine 162 significantly contributes to the NAD⁺ regeneration. This regulation is indispensable for the glycolysis flux and cancer cell proliferation.

5. The text of the manuscript needs major editing. For example, the sentence “Thus, the current data support a model.....glycolytic flux and tumor growth” and the following sentences on page 20 do not make much sense in the context, as stated. References also need correction at several places.

Thanks for raising these issues. We have carefully edited the text including the mentioned sentence. Some of the references have been corrected.

Reviewer #3: Cancer signalling

(Remarks to the Author):

Using cell culture experiments and Xenograft models Cheng et al. examine the involvement of Aurora A in the regulation of glycolysis in cancer cells by phosphorylating Lactate Dehydrogenase B (LDHB). The authors claim that Aurora A phosphorylates LDHB at Serine 162 driving the balance towards the forward reaction that converts pyruvate to lactate. The changes in LDHB enzymatic activity regulate the glycolytic influx and NAD⁺ levels causing a rapid tumor progression in multiple cancers. Furthermore, the expression of Aurora A and LDHB are significantly upregulated in various tumors indicating that both proteins are associated with cancer progression in human patients. Overall the results achieved in this study may be of interest to the field of and could broaden our understanding of metabolism regulation in cancer cells. The manuscript represents an impressive amount of work. However, before publication several key issues need to be addressed:

We appreciate that the reviewer found the novelty and significance of this study, and thank for his/her insightful comments and criticisms

Major points:

1- Figure 2j: The authors claim that S162 is the main phosphorylation site of Aurora A in LDHB based on MS data. It is not clear which peaks represent S162 +/-phosphorylation. Additional experiments need to be included to verify this statement. One of the important papers that has analyzed the consensus site for Aurora A was published by Kettenbach et al. in Science Signaling (28 June 2011). This paper reports the stringent consensus site to be RRXp(S/T) or the less stringent sites to be -2 (RXpS/T) or -3 (RXXp(S/T). S126 as shown in Fig. 2m does not match with all three consensus sites (stringent and non-stringent) that were the result from 778 phosphorylation sites on 562 proteins analyzed by Kettenbach et al.

Therefore, the use of an unspecific pan-phospho Serine antibody to confirm the phosphorylation on S162 by Aurora A is not sufficient. Since the entire paper is based on the claim that Aurora A phosphorylates LDHB at S162 and this phosphorylation regulates glycolysis and tumor growth, there is an urgent need for a phospho-specific antibody that recognizes pS162 as discussed by the authors on page 25.

Thanks for raising this very important issue.

We totally agree with the reviewer that a specific Ab recognizing S162p would be very helpful. Actually, we were aware of this soon after this site was identified in 2015. Therefore we tried to generate such an antibody a couple of times, but neither of them worked. The reason for the failure could be the phosphor-epitope in the antigen (pS162) was veiled by the neighboring cysteine 164 (C164) that was conjugated with KLH, a large (3000+ amino acids) carrier used in the conjugation of the peptide antigen. Several experienced experts in antibody told us that the chance to obtain S162p LDHB antibody is very low. So far, we are not able to obtain the phosphor-LDHB S162 antibody unless new techniques or alternative antigen conjugation methods are available.

As for the consensus motif, a number of well characterized Aurora-A phosphorylated sites do not match the stringent or less stringent sites, such as Histone H3 S10, p53 S215, S315, BRCA1 S308 Therefore, only partial Aurora-A phosphorylated sites match these consensus motifs.

To support the phosphorylation of LDHB S162 by Aurora-A, we now provide more solid data in mass spectrometry analysis. First, the phosphorylation of S162 was detected in endogenous LDHB from multiple cells including DLD1, HeLa and U251 cells (Fig. 2k and Supplementary fig. 2k). Importantly, the phosphorylation of S162, not other sites (S39, S44), was disrupted after Aurora-A depletion, unequivocally demonstrating that Aurora-A

phosphorylates LDHB at S162 in those cells (Fig. 2k and Supplementary fig. 2k-n). In addition, when the recombinant LDHB was phosphorylated in vitro by Aurora-A, S162 was identified to be phosphorylated. Furthermore, Aurora-A overexpression markedly enhanced the pan pSer signal in LDHB wildtype, but not in LDHB S162A expressing cells, suggesting Aurora-A phosphorylates LDHB at serine 162 (Fig. 2l).

Additionally, the enzymatic assays (Fig.3) and functional assays (Fig.5) using LDHB S162A/D mutants also support that Aurora-A regulates LDHB activity via phosphorylating LDHB at S162. For examples, Aurora-A expressing promoted the forward reaction in WT-LDHB expressing cell, but not in LDHB S162A expressing cells (Fig. 3e); Aurora-A inhibition compromised the forward reaction in WT LDHB expressing cells, but not in LDHB S162D expressing cells (Fig. 3g, h); Aurora-A inhibition compromised the glycolysis in WT LDHB expressing cells, but not in LDHB S162D expressing cells (Fig. 5e).

Together, these results demonstrate Aurora-A regulates the activity of LDHB through phosphorylating LDHB at serine 162 in p53 deficient cells.

2- Since the enzymatic activity of Aurora A is cell-cycle regulated with a peak in G2/M, a kinetics of LDHB phosphorylation (S162) in synchronized cells is essential to answer the question whether the phosphorylation of S162 is cell cycle regulated.

According to the data from the Co-IP data and FRET assays, the interaction between Aurora-A and LDHB occurs in interphase, not in mitosis, suggesting the regulation of Aurora-A-LDHB also occurs in interphase, not in mitosis.

In addition, we did not observed any significant phenotypes in mitosis in S162A expressing cells, suggesting LDHB S162 phosphorylation is not required for mitotic progression.

Therefore, the phosphorylation of LDHB S162 by Aurora-A occurs in interphase in p53 deficient tumor cells.

3- Another question that remains unanswered is whether Aurora A is the only protein kinase that might phosphorylate S162. This aspect is very important because the inhibition of Aurora A using MLN8237 shows only a relatively small downregulation of the glycolytic rates (Fig. 1d). The authors used the Aurora A inhibitor MLN8237 at a concentration of 200 nM. At this relatively high concentration the ATP-competitive inhibitor MLN8237 might also inhibit other protein kinases. Therefore, the phosphorylation of S162 should also be tested under Aurora A depletion by RNAi.

This is a very good suggestion. We have done this experiment.

The new data showed that S162 phosphorylation of endogenous LDHB was significantly reduced when Aurora-A was knocked-down in DLD1, Hela and U251 cells (Fig. 2k and supplementary Fig. 2k). Therefore, Aurora-A is the kinase that is responsible for the phosphorylation of LDHB S162 in these cells.

4- A major weakness of the manuscript is that the phosphorylation of S162 was never tested in endogenous LDHB. Expression and phosphorylation of recombinant protein might not reflect the real situation.

The phosphorylation of endogenous LDHB has now been detected in DLD1, HeLa, U251 cells (Fig. 2k and supplementary Fig. 2k).

5- Figure 2. The authors reported the presence of many Aurora A G2/M binding partners (Bora, TPX2) along with LDHB in the IP precipitates. Unless LDHB binds to ARKA during G2/M, this observation is likely due to the fact that these IPs were performed in unsynchronized cells, where the G2/M fraction of cells is somehow enriched. It will be very important to investigate whether the association of Aurora and LDHB is cell cycle - dependent and in which cell cycle phase this interaction takes place. A thymidine synchronization followed by a release of cells should be performed here.

This is also a good suggestion. HeLa cells were synchronized at M phase by releasing from the double thymidine block. The results of the co-IP assay indicated that the association of Aurora-A with LDHB primarily occurs in interphase (Supplementary Fig. 2f). This data was also supported by the FRET assay in which no interaction was detected in mitosis (Supplementary Fig. 2e).

6- Is LDHB regulated in a cell-specific manner?

Base on our data, Aurora-A modulates the activity of LDHB in p53 deficient tumor cells, such as DLD1, HeLa, 293T and U251, but not in p53 competent cells, such as RKO and A549 cells.

7- Suppl. Figure 2. The FRET assays have been performed in randomly taken cells without any synchronization. An analysis in mitotic and non-mitotic cells is required. Single controls are missing (mRuby-Aurora A and single Clover-LDHA)

Very good suggestion. FRET analysis in mitotic cells and single Clover-LDHA have been added (Supplementary Fig. 2e). The results indicate Aurora-A interacts with LDHB in interphase, not in mitosis.

FRET signal of mRuby-Aurora-A alone is too weak to be used for the ratio image generation.

Minor points

8- Figure 2j: The meaning of mass spectrometry data/peaks should be explained in the result part and the legend is too short.

The detailed explanation has been added in the figure legend of Fig.2..

9- Figure 1: For the sake of the general reader more explanation of the ECAR assays should be provided. The figure legends are kept too short.

The detailed explanation has been added in figure legend of Fig.1.

10- Suppl Figure 4c and d: require more explanation within results as well as in legends
*The detailed explanation has been added in both result and legend of Supplementary Fig.4.
A modeled cartoon was added to explain the hydrogen bond between NADH and S162 of LDHB. Thanks for the suggestion.*

11- In the introduction the authors should include a paragraph, where they describe the activation of Aurora A in and outside of mitosis.
The description of Aurora-A activation in interphase and mitosis has been added to the introduction. Thanks for the suggestion.

Thanks again for all the constructive comments and suggestions from all the reviewers.

Sincerely,

Dr. Zhenye Yang

Professor

School of Life Sciences

University of Science & Technology of China, Hefei, China, 230027

Tel: 86-551-63600613

zhenye@ustc.edu.cn

Reviewers' Comments:

Reviewer #1:

Remarks to the Author:

The authors answered to all of my previous comments. I have no additional scientific comment. Supplementary Table 1 does not show up properly in the PDF. To be fixed.

Reviewer #3:

Remarks to the Author:

Despite the presentation of novel data, the claim of S162 being the main phosphorylation site in LDHB by Aurora A is still not convincing. If phospho-specific antibodies for S162 are not available, the evidence has to come from quantitative mass spec.

Major points:

1. The mass spec analysis did not provide solid data or at least the presentation of the data is missing many details. It is unclear how the quantification of phosphorylated peptides derived from endogenous LDHB was performed by mass spec. Specific details should be given in Materials and Methods. Which amino acids were found to be phosphorylated in endogenous LDHB from different cells lines? On page 9 (1. line) the authors point out that different online tools were used and multiple consensus sites for Aurora A in LDHB were found. A schematic figure of the domain structure of LDHB including the consensus sites for the Aurora A and the identified phosphorylation sites in different cells lines should be presented.
2. Figure 2k and Suppl. Fig. 2k: The WB underneath the bar graphs is just showing downregulation of Aurora A using shRNA in DLD and HeLa cells. The reduction of S162 phosphorylation is depicted as bar graphs. Since the author do not have a specific antibody against pS162, how can they claim that this downregulation can only be ascribed to the reduction of phosphorylation of S162. In order to demonstrate this, the authors need to knockdown endogenous LDHB and replace it with exogenous fusion protein carrying a non-phosphorylatable mutant of S162 (pS162A) and the phosphor-mimetic S162E (a WT construct should also be considered. It will serve as control). Afterwards the author should check on the phosphorylation status of the LDHB fusion constructs in the presence and absence of Aurora A. Here a pan phospho-antibody against pSer might be used to detect the changes in the phosphorylation status of both exogenously expressed mutants.
3. Figure 2l: It is not very likely that Aurora A phosphorylates only one site in LDHB or in other words that Aurora A ignores all consensus sites in LDHB, but instead phosphorylates only one unconventional site. The use of an pSer-antibody is not very helpful for an "unequivocal" demonstration of a phosphorylation site in this in vitro experiment. Mass spec. needs to be done to show that LDHB is phosphorylated and which sites are phosphorylated in vitro.
4. The main kinase activity of Aurora A can be measured in G2/M. I have still doubts that Aurora A phosphorylates LDHB in interphase. A cell cycle analysis was still not performed. The authors need to measure Aurora A activity in double thymidine- treated and synchronized cells. The kinetics of Aurora A activity along the cell cycle needs to be performed including Aurora A autophosphorylation and endogenous substrates of Aurora A using phospho-antibodies that are commercially available. FACS measurements need to show the distribution of cells in cell cycle phases.
5. The legends still do not contain experimental details. As an example: Legend of Suppl. Fig. 2f. The method for arresting cells in G2/M is missing. A FACS control is missing.

6. My question whether LDHB is cell-cycle regulated is still not answered.

7. The authors claim in the Introduction that Aurora A is phosphorylated at Thr288 and activated in interphase. Publications that support this statement should be cited accompanied by experimental data showing the Aurora A is active in interphase.

Reviewer #4:

Remarks to the Author:

In their manuscript "Aurora-A Mediated Phosphorylation of Lactate Dehydrogenase B promotes glycolysis and tumor growth by relieving the substrate inhibition effect" Cheng et al. investigate the role of Aurora A phosphorylation on lactate dehydrogenase B (LDHB) activity. They show that Aurora A phosphorylation of LDHB relieves substrate inhibition promoting the reduction of pyruvate to lactate. This is an interesting study. While the analysis of the effects of S162 phosphorylation on LDHB is well carried out, the link with Aurora A and p53 are weak and requires clarification and additional experimental validation before publication.

1. A major concern is that all experiments carried out with the Aurora A inhibitor MLN8237 are performed for 4-6 hrs. Aurora A activation occurs through autophosphorylation and 45min to 1hr treatments are sufficient to affect downstream phosphorylation. The prolonged exposure of cells to MLN8237 is problematic and leads to indirect effects. Did the authors try shorter treatments without success? Or have they never tried shorter treatments?

2. The identification of LDHB as an Aurora A interacting protein by mass spectrometry was carried out with control. They identified about 500 proteins with higher confidence than LDHB as Aurora A interactors. In fact, in addition to LDHB, the authors identify other glycolysis enzymes (PKM, PGAM1, ENOA, G3P, etc.) in their MS experiment but do not comment on them. Glycolytic enzymes are abundant and often unspecific binders in interactomes MS experiments. This makes this a weak identification. While the authors validate the interaction, the current description of its identification by MS is misleading and should be adjusted.

3. The authors predict potential Aurora A phosphorylation sites on LDHB. Using MS, the authors identify S162 as an Aurora A phosphorylation site. The phosphorylation prediction statement is misleading because S162 does not correspond to the canonical Aurora A motif and cannot be predicted, which should be noted in the manuscript. The MS data for all identified sites should be included as a supporting table. The authors also indicate that they identify the site from DLD1, 293T, HeLa and U251 cell lines, this information should be included.

4. The authors validate S162 as an Aurora A phosphorylation by in vitro kinase assay, MLN8237 inhibitor treatment, and Aurora A depletion. Because MLN8237 treatment and Aurora A depletion are carried out for a prolonged period of time, they do not prove a direct kinase – substrate relationship. The authors should analyze the in vitro kinase assay by MS to support this claim. The spectra in Fig 2J needs to be annotated to be informative for the reader or removed. The authors also identify other sites that do not change upon prolonged treatment. A table with the identified peptides, parameters of the identification, and peptide abundance needs to be included as a supplemental table. A description of the quantification of the MS results for the phosphorylation sites should be included. Did the authors calculate occupancy or relative differences? The error bars are missing from Supp 2K, Supp 2N, and Supp 2O. The authors state that S162 is significantly increased upon Aurora A overexpression (Supp 2N). There is no quantification and no statically test performed to support this statement, thus, "significantly" should be removed.

5. The authors state that the regulation of glycolysis by Aurora A is linked to p53 status. To make the p53 claim, the authors should manipulate p53 levels. For instance, the authors could deplete p53 in DLD1, U251, or 293T cells or stably express p53 in A549, MCF7 or RKO cells and show the

effects on Aurora A levels and activity, and glycolysis.

Minor comments:

1. The authors state that Aurora A and LDHB overexpression is associated with cancer progression in human patients. However, the authors only compare normal, and tumor samples not tumor progression. Thus cancer progression is not evaluated. This should be corrected in the text.
2. Chemotrypsin is misspelled
3. The manuscript needs editing to help with comprehension and clarity especially at the beginning of the manuscript; the later part are more comprehensible.

Reviewer #5:

Remarks to the Author:

The authors have adequately addressed the concerns raised by the second reviewer.

Point-by-Point Response to Reviewers' Comments

We are very grateful for these insightful criticisms and constructive comments from the reviewers. Now we have addressed all these issues by doing further work, adding more detailed information and rephrasing the interpretation, which substantially support the conclusions and improve the overall quality of the manuscript.

Listed below are the point-by-point responses (**blue font**) to the reviewers' critiques. All the changes in the manuscript are highlighted as **red font**.

Reviewer#3

Despite the presentation of novel data, the claim of S162 being the main phosphorylation site in LDHB by Aurora A is still not convincing. If phospho-specific antibodies for S162 are not available, the evidence has to come from quantitative mass spec.

We agree with the reviewer that quantitative mass spectrometry (MS) analysis is critical for validating the phosphorylation site in this study. Indeed, we have performed a series of MS analyses (25+ tests) during the original study and the revision. However, as the reviewer pointed out, we did not clearly present the results and the evidence is not sufficient. Therefore, to convincingly address this issue, the following work has been done following the reviewer's suggestion:

1. Data of the MS analysis from multiple in vitro and in vivo assays were organized in one file with multiple worksheets (Supplementary Table 2 in the new version).
2. Raw MS data has also been supplemented for reference (Supplementary Table 3)
3. The details of the data (page 9-10) and detailed methods for quantitation of MS data (page 46-47) have been supplemented.
4. The phosphorylation of LDHB in cancer cells were also analyzed by MS in S and M phases of the cell cycle, two phases in which the associations of LDHB and Aurora-A

are different (Fig. 2e and supplementary Fig. 2n). While the proportion of S162 phosphorylation is more than 20% in S phase, the proportion is less than 2% in M phase. Therefore, Both the association of Aurora-A with LDHB and the highest proportion of S162 phosphorylation occurred in S phase of cell cycle, suggesting Aurora-A phosphorylates LDHB at S162

5. Endogenous LDHB was depleted and the shRNA resistance WT/S162A was expressed. Aurora-A overexpression enhanced the total level of phosph-ser in wildtype but not S162A expression cells (Fig. 2l), suggesting S162 is the main Aurora-A phosphorylation site on LDHB.

In addition, the following evidence in the last version of the manuscript supports that S162 is the primary Aurora-A phosphorylation site on LDHB in Aurora-A overexpressing cancer cells:

1. The proportion of S162 phosphorylation was significantly reduced from 30% and 22% to 6.6% and 0 after Aurora-A knocked-down in DLD1 and Hela cells respectively (Fig. 2k), whereas the proportion of S39 and S44 phosphorylation remained unchanged (less than 2%). These data indicated that Aurora-A phosphorylates LDHB at S162.
2. The phosphorylation of LDHB S162 was decreased when Aurora-A is inhibited, and the phosphorylation of S162 was increased when cells were cultured in hypoxia condition (Supplementary Table 2). Their interaction was also enhanced under hypoxia (Fig.2f and supplementary Fig. 2l).
3. Aurora-A expression enhanced the enzymatic activity of wild type LDHB, but not S162A (Fig. 3e). Aurora-A inhibition reduced the enzymatic activity of LDHB, but not S162D (Fig. 3h).

Taken together, from these evidence, we concluded that Aurora-A primarily phosphorylates LDHB at S162 in Aurora-A overexpressing cancer cells. Remarkably, in the metabolism assays, Aurora-A inhibition led to significant reductions in glycolytic rates and glycolytic

intermediates levels in wildtype LDHB, but not S162D expressing cells (Fig. 5), supporting Aurora-A mediated phosphorylation of LDHB S162 is mainly responsible for the regulation of LDHB in cancer cells.

Major points:

1. The mass spec analysis did not provide solid data or at least the presentation of the data is missing many details. It is unclear how the quantification of phosphorylated peptides derived from endogenous LDHB was performed by mass spec. Specific details should be given in Materials and Methods. Which amino acids were found to be phosphorylated in endogenous LDHB from different cells lines? On page 9 (1. line) the authors point out that different online tools were used and multiple consensus sites for Aurora A in LDHB were found. A schematic figure of the domain structure of LDHB including the consensus sites for the Aurora A and the identified phosphorylation sites in different cells lines should be presented.

We appreciate the reviewer for these constructive criticisms. We have now provided much more detailed information in Methods (page 46-47) and Results sections (page 9-10)

For the details of MS data from in-vitro and in vivo assays, we have re-arranged the data into an excel file (Supplementary Table 2) with seven worksheets showing statistical results of MS analyses for LDHB phosphorylation. Furthermore, all raw MS data containing identified peptides, parameters of the identification etc. was supplemented (Supplementary Table 3).

Briefly, S70 (proportion of phosphorylation: 0.53%), S85 (1.7%), S162 (8.9%), T302 (7.3%), S303 (6.1%), S320 (3.5%) and T323 (0.5%) were detected to be phosphorylated in vitro. However, in DLD1, HeLa, U251 and 293T cells, three phosphorylated sites were identified: S162 (22-30%), S44 (0-2.3%), S39 (0-2.3%). S162 is the only site to be phosphorylated in both in-vitro assay and in cancer cells. Moreover, S162 is the only site

that can respond to Aurora-A deletion or inhibition. The inconsistency for other phosphorylation sites between in-vitro and in vivo assay may be ascribed to the difference in tetramer formation (LDHA+LDHB in vivo; LDHB only in vitro) or other unknown regulations in cultured cells.

We agree that S162 is not a canonical site for Aurora kinase, and as the other reviewer also pointed out, the statement of prediction of the phosphorylation is misleading. Therefore, the statement has been removed and the description has been modified accordingly.

2. Figure 2k and Suppl. Fig. 2k: The WB underneath the bar graphs is just showing downregulation of Aurora A using shRNA in DLD and HeLa cells. The reduction of S162 phosphorylation is depicted as bar graphs. Since the author do not have a specific antibody against pS162, how can they claim that this downregulation can only be ascribed to the reduction of phosphorylation of S162.

This was a misunderstanding. The percent of pS162 was quantified using data from MS analysis, not from the immunoblotting data using pSer antibody. As the reviewer commented, the data wasn't clearly presented. Now we have modified the presentation of this data. Therefore, the bar graphs showed that after Aurora-A knockdown, the proportion of S162 phosphorylation was significantly reduced in DLD1 and HeLa cells.

In order to demonstrate this, the authors need to knockdown endogenous LDHB and replace it with exogenous fusion protein carrying a non-phosphorylatable mutant of S162 (pS162A) and the phosphor-mimetic S162E (a WT construct should also be considered. It will serve as control). Afterwards the author should check on the phosphorylation status of the LDHB fusion constructs in the presence and absence of Aurora A. Here a pan phospho-antibody against pSer might be used to detect the changes in the phosphorylation status of both exogenously expressed mutants.

This is a very good idea. Following the reviewer's suggestion, the endogenous LDHB was not knocked down and shRNA-resistant LDHB WT and S162A mutant were expressed with/without Aurora-A overexpression in DLD1 cells. Pan phosphor-Ser blot showed that Aurora-A expression increased the level of pSer in wildtype LDHB but not in S162A mutant, suggesting that S162 is the main Aurora-A-phosphorylated serine in LDHB (Fig. 2l).

3. Figure 2l: It is not very likely that Aurora A phosphorylates only one site in LDHB or in other words that Aurora A ignores all consensus sites in LDHB, but instead phosphorylates only one unconventional site. The use of an pSer-antibody is not very helpful for an "unequivocal" demonstration of a phosphorylation site in this in vitro experiment. Mass spec. needs to be done to show that LDHB is phosphorylated and which sites are phosphorylated in vitro.

We had performed a MS analysis to identify phosphorylation of LDHB site after incubated with Aurora-A in-vitro. Seven residues, S70 (proportion of phosphorylation: 0.53%), S85 (1.7%), S162 (8.9%), T302 (7.3%), S303 (6.1%), S320 (3.5%) and T323 (0.5%) were detected to be phosphorylated in vitro. However, in DLD1, HeLa, U251 and 293T cells, three phosphorylated sites were identified: S162 (%), S44 (%), S39 (%). S162 is the only site to be phosphorylated in both in-vitro assay and in cancer cells. Moreover, S162 is the only site that can respond to Aurora-A deletion or inhibition. The inconsistency of other phosphorylation sites between in-vitro and in vivo assay may be ascribed to the difference in tetramer formation (LDHA+LDHB in vivo; LDHB only in vitro) or other unknown regulations in cultured cells.

Also see the responses to the first two comments above.

4. The main kinase activity of Aurora A can be measured in G2/M. I have still doubts that Aurora A phosphorylates LDHB in interphase. A cell cycle analysis was still not performed. The authors need to measure Aurora A activity in double thymidine- treated and synchronized cells. The kinetics of Aurora A activity along the cell cycle needs to be performed including Aurora A autophosphorylation and endogenous substrates of Aurora A using phospho-antibodies that are commercially available. FACS measurements need to show the distribution of cells in cell cycle phases.

Although Aurora-A has the highest kinase activity in G2 and M phase, it has been reported to be activated during interphase in cancer cells (also see the response to major issue 7 below). To assess Aurora-A activity in interphase in our experiment, HeLa cells were synchronized by double thymidine treatment as suggested by the reviewer. FACS data (Supplementary Fig. 2k) showed that cells were well synchronized at G1, S, G2 and M phases. The blot of cyclin-B (CCNB1) could also serve as a marker for cell cycle stage (Fig. 2e). We did not find a commercially available phosphor-antibody for Aurora-A substrates in interphase cells, therefore the activity of Aurora-A was assessed by Aurora-A T288p antibody. As expected, the level of Aurora-A T288p is high in M phase and very low in G1 phase (Fig. 2e). Notably, Aurora-A showed modest activity in S phase (Fig. 2e). Co-IP data indicated that the interaction between Aurora-A and LDHB is strongest in S phase and modest in G2 phase, but their interaction is weak in G1 and M phase. Notably, the proportion of S162 phosphorylation is also much higher in S phase than in M phase (Supplementary Fig. 2n). Thus, these data suggest that Aurora-A mainly phosphorylates LDHB in S and G2 phases.

5. The legends still do not contain experimental details. As an example: Legend of Suppl. Fig. 2f. The method for arresting cells in G2/M is missing. A FACS control is missing. We have added the experimental details in the Methods section and figure legends. The

FACS control has been included (supplementary Fig. 2k).

6. My question whether LDHB is cell-cycle regulated is still not answered.

To address this question, HeLa cells were synchronized at G1, S, G2 and M phases (Supplementary Fig. 2k, detailed method in page 47-48). The interactions between LDHB and Aurora-A at different cell cycle phases were examined by co-IP and FRET. Interestingly, the interaction is strongest in S phase and modest in G2 phase, while their interaction is very weak in M and G1 phases (Fig. 2e and supplementary Fig. 2e), indicating Aurora-A primarily phosphorylates LDHB at S phase. The phosphorylation of LDHB S162 was also quantified by MS analysis in cells at S phase and M phase. Consistently, the frequency of S162 phosphorylation in S phase is much higher than that in M phase (Supplementary Fig. 2n, Table S2). Moreover, LDHB S162A expressing cell displayed S phase delay (Supplementary Fig. 6a, b), suggesting that phosphorylation of LDHB S162 mainly occurs at S phase.

Together, these data indicated that the regulation of LDHB by Aurora-A mainly occurs in S and G2 phase, stages in which glycolysis and bio-synthesis are highly demanded in cancer cells.

7. The authors claim in the Introduction that Aurora A is phosphorylated at Thr288 and activated in interphase. Publications that support this statement should be cited accompanied by experimental data showing the Aurora A is active in interphase.

It had been reported that activated Aurora-A phosphorylates various substrates in interphase of the cell cycle in a number of studies. We have cited two studies in which Aurora-A was proved to be activated in interphase by using the FRET sensor, pHT288 and pHH3 (phosphor-histone H3) antibodies.

The data in these two studies are attached below and the statements with Aurora-A activation in interphase have been highlighted.

In our study, we demonstrated that Aurora-A can be activated (T288ph) in S and G2 of the interphase in cancer cells (Fig. 2e and supplementary Fig. 2e).

1. Bertolin G, Sizaire F, Herbomel G, Rebutier D, Prigent C, Tramier M. A FRET biosensor reveals spatiotemporal activation and functions of aurora kinase A in living cells. *Nat Commun* 2016;7:12674

AURKA-specific siRNA. $n = 100$ cells per condition scored in each of three independent experiments. (b) Representative fluorescence (GFP channel) and corresponding lifetime images of GFP-AURKA and GFP-AURKA-mCherry U2OS stable cell lines not synchronized (left panels) or synchronized in mitosis (right panels) and illustrating the presence of both proteins at the centrosome, in the cytosol and at mitotic spindles. Graphs: corresponding quantification of the lifetime of EGFP in the two cell lines and in the indicated subcellular compartments. Centrosomes were labelled with CETN1-IRFP670. $n = 30-40$ cells per condition from three independent experiments. (c) Quantification of the lifetime of EGFP GFP-AURKA and GFP-AURKA-mCherry cells synchronized at mitosis and treated with dimethylsulfoxide (DMSO) or with MLN8237. $n = 30-40$ cells per condition from three independent experiments.

ARTICLE

NATURE COMMUNICATIONS | DOI: 10.1038/ncomms12674

AURKA-mCherry showed a mean lifetime comparable to the one of GFP-AURKA alone throughout the treatments, thus not indicative of intermolecular FRET between two distinct donor-acceptor molecules.

As the AURKA biosensor is phosphorylated when purified from *E.coli*, we evaluated whether the incubation of GFP-AURKA-mCherry with increasing quantities of untagged AURKA could decrease FRET between EGFP and mCherry, indicating intermolecular FRET between neighbouring proteins (Fig. 3b). The lifetime of the biosensor remained unaltered for all

GFP-AURKA-mCherry (Fig. 1b). In interphase cells FRET could be detected in the cytosol as well, although at lower levels (Fig. 4b). To further correlate FRET and the activation of AURKA, we analysed the inhibitory effect of MLN8237 on mitotic cells, in which the kinase activity of AURKA is greater compared with other cell cycle phases⁴². No significant FRET was measured on the mitotic spindle of cells treated with MLN8237 compared with controls (Fig. 4c). We also analysed the conformational changes of AURKA in the presence of the Lys162Met kinase-dead variant. We stably expressed this

In this study, the authors developed a TRET sensor that detects Aurora-A activation. They reported that Aurora-A is also active in interphase cells (highlighted text above).

2. Plotnikova OV, Pugacheva EN, Dunbrack RL, Golemis EA. Rapid calcium-dependent activation of Aurora-A kinase. *Nat Commun* 2010;1:64

In figure 1 and 2, the authors showed Aurora-A could be activated after the release of calcium in interphase. The activation of Aurora-A was proved by phT288 and phHH3 antibodies

Figure 1 | Calcium release rapidly induces AurA auto-phosphorylation. (a) Western blot assay confirming expression of the AVPR1 receptor in cell lines used in this study. MCF7 were used as a positive control for AVPR1 expression, based on North et al.⁶⁰. (b) Immunofluorescence of HK-2 cells 30 s after stimulation with 100 nM arginine vasopressin (+ AVP) versus control untreated cells (- AVP). Cells were visualized with antibodies to AurA (red) and T²⁸⁸-phospho-AurA (phAurA, green) and DAPI (blue), as indicated; scale bar, 20 μ m. Insets show magnification of indicated centrosomes. (c) Analysis of data from b quantifies phAurA-positive cells at times after stimulation. (d) Analysis of data from b quantifies relative intensity of AurA in cells within 20 s after stimulation with AVP. (e) Western blot analysis of AVP-treated HEK293 cells overexpressing AurA in the presence or absence of extracellular Ca²⁺ chelator (5 mM EGTA) (f) Western blot analysis of phAurA in HEK293 cells transfected with an AurA-expressing plasmid 1-2 min after stimulation with 10 μ M histamine. For each SDS-PAGE, β -actin is used as a loading control and molecular weight is indicated in kDa. For c and d, an average of 150 cells were counted. (c, d, e) N = 3 with error bars indicating the standard error. *P < 0.05.

Figure 2 | Intracellular calcium release is sufficient for AurA activation. (a) HEK293 cells overexpressing AurA were incubated with 5 μ M thapsigargin for the indicated periods of time, and then cell lysates were analysed by western blot. Graph shown below indicates ratio of phAurA to total AurA. (b) AurA immunoprecipitation and *in vitro* kinase assay, after thapsigargin treatment. HEK293 cells overexpressing AurA were treated with 5 μ M thapsigargin or control dimethylsulfoxide (DMSO) for 5 min, AurA was immunoprecipitated and incubated with histone H3 substrate in an *in vitro* kinase assay. pHHH3, antibody to phosphorylated histone H3. * $P < 0.05$. (c) Left, an experiment similar to that shown in a, except that it is performed in the presence of 50 μ M BAPTA-AM or control DMSO. Right, HEK293 cells overexpressing AurA were incubated with 20 μ M BAPTA-AM (30 min) plus 7.5 mM EGTA (2 min) before the addition of 5 μ M thapsigargin for the indicated periods of time, with analysis as for a. (d) An experiment similar to that shown in a, except that it is performed in the presence of 5 mM EGTA alone for the indicated periods of time (right panel) or in the presence of 5 mM EGTA (2 min) before the addition of 5 μ M thapsigargin. (e) HEK293 cells overexpressing AurA were incubated with 5 μ M thapsigargin and indicated concentration of ionomycin for the indicated periods of time, and then cell lysates were analysed by western blot using indicated antibodies. (f) HK2 cells were treated with DMSO, 5 μ M thapsigargin (Tg) or 0.5 μ M ionomycin (Ion), and then T²⁸⁸-phAurA levels were analysed by enzyme-linked immunosorbent assay (ELISA), based on A_{450nm}. To confirm the specificity of ELISA signal, a parallel group of HK2 cells were pre-treated with the AurA inhibitor PHA680632 (PHA) at 500 nM concentration. * $P < 0.05$. Lysis buffer was used as a negative control. (g) AurA-overexpressing HEK293 cells were transfected with siRNA-depleting NEDD9 (siNEDD9) or scrambled control siRNA (scr), and then were incubated with 5- μ M thapsigargin and processed as in a. For this and subsequent SDS-PAGE analyses, each experiment was performed three times independently, with error bars indicating the standard error. * $P < 0.05$.

Reviewer#4

In their manuscript “Aurora-A Mediated Phosphorylation of Lactate Dehydrogenase B promotes glycolysis and tumor growth by relieving the substrate inhibition effect” Cheng et al. investigate the role of Aurora A phosphorylation on lactate dehydrogenase B (LDHB) activity. They show that Aurora A phosphorylation of LDHB relieves substrate inhibition promoting the reduction of pyruvate to lactate. This is an interesting study. While the analysis of the effects of S162 phosphorylation on LDHB is well carried out, the link with Aurora A and p53 are weak and requires clarification and additional experimental validation before publication.

We thanks the reviewer for well summarizing our manuscript.

1. A major concern is that all experiments carried out with the Aurora A inhibitor MLN8237 are performed for 4-6 hrs. Aurora A activation occurs through autophosphorylation and 45min to 1hr treatments are sufficient to affect downstream phosphorylation. The prolonged exposure of cells to MLN8237 is problematic and leads to indirect effects. Did the authors try shorter treatments without success? Or have they never tried shorter treatments?

This is a reasonable concern. To address this issue, we have tested shorter MLN8237 treatment in multiple assays. First, Aurora-A T288-ph and glycolytic levels were measured in cells treated with MLN8237 for 1, 2 and 4 hours (Fig. 1C). Indeed, one hour treatment is sufficient to inhibit Aurora-A. Similarly, one hour treatment significantly reduced the glycolytic rates though the reduction of glycolytic level in one hour treatment is less than that in cells treated for four hours. Second, the interaction between Aurora-A and LDHB was assessed after MLN8237 treatment for 1, 2, 4 hours (Fig. 2g). Similarly, one hour treatment is enough to markedly reduce their interaction. Further, using NADH/NAD+

sensor, enzymatic activity of LDHB was measured in live cell after MLN8237 treatment for 1 and 2 hours (Supplementary Fig. 3f). Consistently, one hour treatment was sufficient to significantly inhibit LDH activity.

Therefore, these data suggested that the inhibition of glycolysis, the reduction of Aurora-A-LDHB interaction and the decrease of LDH activity are direct effects of Aurora-A inactivation.

2. The identification of LDHB as an Aurora A interacting protein by mass spectrometry was carried out with control. They identified about 500 proteins with higher confidence than LDHB as Aurora A interactors. In fact, in addition to LDHB, the authors identify other glycolysis enzymes (PKM, PGAM1, ENOA, G3P, etc.) in their MS experiment but do not comment on them. Glycolytic enzymes are abundant and often unspecific binders in interactomes MS experiments. This makes this a weak identification. While the authors validate the interaction, the current description of its identification by MS is misleading and should be adjusted.

Thanks very much for raising this issue. We had noticed that other glycolytic enzymes were also identified in the precipitant. We are particularly interested in LDHB is because LDH is the enzyme responsible for the lactate production and NAD⁺ regeneration, which modulate the glycolytic rate. Moreover, LDHA has been proved to be a key target for aerobic glycolysis, however, how LDHB is regulated is still elusive. Therefore, we wondered whether Aurora-A regulate glycolysis through specifically interacting with LDHB, not LDHA (more abundant).

Following the reviewer's suggestion, the description of MS results has been elaborated accordingly. Thanks very much for the suggestion.

3. The authors predict potential Aurora A phosphorylation sites on LDHB. Using MS, the authors identify S162 as an Aurora A phosphorylation site. The phosphorylation prediction statement is misleading because S162 does not correspond to the canonical Aurora A motif and cannot be predicted, which should be noted in the manuscript. The MS data for all identified sites should be included as a supporting table. The authors also indicate that they identify the site from DLD1, 293T, HeLa and U251 cell lines, this information should be included.

These are very good suggestions. The MS data for all identified sites from in vitro assay and various cell lines has been provided in the Supplementary table 2. We agree with the reviewer that the prediction for S162 phosphorylation is misleading, therefore the statement has been removed. The description of phosphorylation site has been included accordingly.

4. The authors validate S162 as an Aurora A phosphorylation by in vitro kinase assay, MLN8237 inhibitor treatment, and Aurora A depletion. Because MLN8237 treatment and Aurora A depletion are carried out for a prolonged period of time, they do not prove a direct kinase – substrate relationship. The authors should analyze the in vitro kinase assay by MS to support this claim. The spectra in Fig 2J needs to be annotated to be informative for the reader or removed. The authors also identify other sites that do not change upon prolonged treatment. A table with the identified peptides, parameters of the identification, and peptide abundance needs to be included as a supplemental table. A description of the quantification of the MS results for the phosphorylation sites should be included. Did the authors calculate occupancy or relative differences? The error bars are missing from Supp 2K, Supp 2N, and Supp 2O. The authors state that S162 is significantly increased upon Aurora A overexpression (Supp 2N). There is no quantification and no statically test performed to support this statement, thus, “significantly” should be removed.

We thank the reviewer for pointing out these issues. Below we will address them one by one.

- His-tagged LDHB was subjected to the MS analysis after in vitro kinase assay with recombinant Aurora-A (Supplementary Table2). Seven residues including S162 were identified to be phosphorylated by Aurora-A in vitro. S162 has the highest ratio to be phosphorylated among these sites. More importantly, S162 was the only one that was also identified in cell lines. Thus, these MS data indicate that Aurora-A phosphorylates LDHB at S162. Also see the above response to first comment from reviewer #3.
- The MS data (Fig. 2j) has been regenerated and annotated.
- The detailed information of phosphorylated sites and peptides has been included in Supplementary Table3. All potential phosphorylation sites of Aurora-A kinase except S269, a non-conserved residue, were covered in the MS analysis.
- Instead of showing the bar graph, the MS data for Supp 2K, Supp 2N, and Supp 2O has now been moved to supplementary Table 2. A new bar graph with statistical error bars for the MS analyses in S or M phase of the cell cycle has been shown. The statement has been modified accordingly.

5. The authors state that the regulation of glycolysis by Aurora A is linked to p53 status. To make the p53 claim, the authors should manipulate p53 levels. For instance, the authors could deplete p53 in DLD1, U251, or 293T cells or stably express p53 in A549, MCF7 or RKO cells and show the effects on Aurora A levels and activity, and glycolysis.

Following the reviewer's suggestion, p53 activity was disrupted by overexpressing a dominant negative mutant R273H in A549 cells. Indeed, Aurora-A was upregulated and activated after p53 inactivation by R273H expressing. Consistently, the level of glycolysis was significantly increased in R273H expressing A549 cells (Fig. 1h). By contrast, this increase was mostly abolished after Aurora-A inhibition (Fig. 1h), suggesting activated

Aurora-A is required for the upregulation of glycolysis in p53 deficient cells. Furthermore, the p53 isogenic HCT116 cells were used to assess the contribution of p53 in Aurora-A expression and glycolysis. Notably, in p53 null cells the glycolysis and the activity of Aurora-A were significantly increased, however, this increase is largely abolished after Aurora-A inhibition (supplementary fig. 1i).

Together, these data demonstrated that the regulation of glycolysis by Aurora-A is link to p53 status.

Minor comments:

1. The authors state that Aurora A and LDHB overexpression is associated with cancer progression in human patients. However, the authors only compare normal, and tumor samples not tumor progression. Thus cancer progression is not evaluated. This should be corrected in the text.

We have modified the statement according to the suggestion. Thanks very much.

2. Chemotrypsin is misspelled

It has been corrected. Thanks.

3. The manuscript needs editing to help with comprehension and clarity especially at the beginning of the manuscript; the later part are more comprehensible.

The beginning of the manuscript has been extensively edited to help with comprehension.

Reviewers' Comments:

Reviewer #3:

Remarks to the Author:

My main concern focused on the confirmation of Ser162 as the major phosphorylation site on LDHB, especially when knowing that this serine does not represent a canonical site for Aurora A. Even if the authors were not able to rise specific phospho-antibodies that could validate in vivo phosphorylated LDHB at Ser162, they presented convincing MS data (from in vitro and in vivo experiments) that recognized a number of putative sites in LDHB. More importantly, MS data identified Ser162 as the major phosphorylation site of Aurora A. Furthermore, replacement experiments, in a LDHB-downregulated background, using the non-phosphorylatable mutant S162A showed a reduction of the overall phosphorylation status of LDHB, which may represent an additional indicator that Ser162 is indeed the targeted site of Aurora A on LDHB.

The authors presented as requested citations/publications supporting their findings regarding the interphase activity of Aurora-A. The authors were able to show the presence of phospho-T288 (ARKA auto-phosphorylation) signal during S-phase of synchronized cells, even though, T288 signal was much weaker as in G2-phase (Fig. 2e).

The main text has also been improved. Experimental details have been introduced in the material and methods section.

In summary, all our concerns have been addressed adequately. Therefore, I encourage the journal to publish the revised version.

Reviewer #4:

Remarks to the Author:

In their revision, the authors address most of my concerns. As requested, the authors provide additional information on how phosphorylation sites were quantified.

This was done incorrectly. The added Table S2 is misleading and incorrect and needs to be replaced. The authors quantify the percentage of phosphorylation of S162 and other sites by dividing the number of phosphorylated peptides by the total number of peptides with the same backbone sequence. This is not appropriate. The phosphorylated and unphosphorylated form of the same backbone peptide sequence can behave differently during analysis and cannot be directly compared. Supp table 2 is not informative and misleading. All data and conclusions based on these analyses must be removed and corrected. This includes at least Supp Table 2, Fig 2K, Supp Fig 2N.

To determine occupancy the authors should dephosphorylate peptides after digestion and determine the abundance (peak area not peptide counts) of the unphosphorylated peptide before and after dephosphorylation to determine the increase in abundance due to dephosphorylation (see Wu et al Nat Methods 2011 PMID:21725298). Alternatively, to get a relative sense of the difference in abundance, the phosphopeptide peak area could be compared between conditions. Here the authors should correct for LDHB abundance differences between these conditions.

This needs to be addressed before this manuscript can be published.

Point-by-Point Response to Reviewers' Comments

We are also very grateful for the insightful criticism on the quantitative MS data from the reviewer #4. The quantitation of the MS data has been re-analyzed following the reviewer's suggestion.

Listed below are the point-by-point responses (**blue font**) to the reviewers' critiques.

The changes in the manuscript are highlighted as **red font**.

Reviewer #4 (Remarks to the Author):

In their revision, the authors address most of my concerns. As requested, the authors provide additional information on how phosphorylation sites were quantified.

This was done incorrectly. The added Table S2 is misleading and incorrect and needs to be replaced. The authors quantify the percentage of phosphorylation of S162 and other sites by dividing the number of phosphorylated peptides by the total number of peptides with the same backbone sequence. This is not appropriate. The phosphorylated and unphosphorylated form of the same backbone peptide sequence can behave differently during analysis and cannot be directly compared. Supp table 2 is not informative and misleading. All data and conclusions based on these analyses must be removed and corrected. This includes at least Supp Table 2, Fig 2K, Supp Fig 2N.

We appreciate the reviewer for pointing out the errors in quantitative MS of phosphorylated peptide. After extensive discussions, we now fully understood these concerns and agreed with the reviewer on this point. The quantitation has been performed following the reviewer's suggestion and methods in the references (Wu et al Nat Methods 2011¹; Wu et al Mol Cell Proteomics 2011²; Steen H et al PNAS 2005³).

1 Wu, R. *et al.* A large-scale method to measure absolute protein phosphorylation stoichiometries. *Nature methods* **8**, 677-683, doi:10.1038/nmeth.1636 (2011).

2 Wu, R. *et al.* Correct interpretation of comprehensive phosphorylation dynamics requires normalization by protein expression changes. *Molecular & cellular proteomics : MCP* **10**, M111

009654, doi:10.1074/mcp.M111.009654 (2011).

- 3 Steen, H., Jebanathirajah, J. A., Springer, M. & Kirschner, M. W. Stable isotope-free relative and absolute quantitation of protein phosphorylation stoichiometry by MS. *Proc Natl Acad Sci U S A* **102**, 3948-3953, doi:10.1073/pnas.0409536102 (2005).

To determine occupancy the authors should dephosphorylate peptides after digestion and determine the abundance (peak area not peptide counts) of the unphosphorylated peptide before and after dephosphorylation to determine the increase in abundance due to dephosphorylation (see Wu et al Nat Methods 2011 PMID:21725298). Alternatively, to get a relative sense of the difference in abundance, the phosphopeptide peak area could be compared between conditions. Here the authors should correct for LDHB abundance differences between these conditions.

This needs to be addressed before this manuscript can be published.

To quantify the relative abundance of LDHB phosphopeptide containing S162, the phosphorylation position was validated using the ptmRS algorithm. Phosphopeptide abundances were further normalized against the abundance of total peptides for LDHB. The relative abundance was then compared between different conditions.

Accordingly, the figure 2k, supplementary figure 2n and supplementary Table 2 have been replaced.

Thanks again for pointing out this issue.

Reviewers' Comments:

Reviewer #4:

Remarks to the Author:

The authors have fully addressed my concern.